# Liquid metal droplets bouncing higher on thicker water layer

Yuhang Dai[1,2,5], Minfei Li[1,5], Bingqiang Ji [1,5], Xiong Wang[1], Siyan Yang[1], Peng Yu [2], Steven Wang[1] ✉, Chonglei Hao [3] ✉ & Zuankai Wang [1,4] ✉

Liquid metal (LM) has gained increasing attention for a wide range of applications, such as flexible electronics, soft robots, and chip cooling devices, owing to its low melting temperature, good flexibility, and high electrical and thermal conductivity. In ambient conditions, LM is susceptible to the coverage of a thin oxide layer, resulting in unwanted adhesion with underlying substrates that undercuts its originally high mobility. Here, we discover an unusual phenomenon characterized by the complete rebound of LM droplets from the water layer with negligible adhesion. More counterintuitively, the restitution coefficient, defined as the ratio between the droplet velocities after and before impact, increases with water layer thickness. We reveal that the complete rebound of LM droplets originates from the trapping of a thinly low-viscosity water lubrication film that prevents droplet-solid contact with low viscous dissipation, and the restitution coefficient is modulated by the negative capillary pressure in the lubrication film as a result of the spontaneous spreading of water on the LM droplet. Our findings advance the fundamental understanding of complex fluids' droplet dynamics and provide insights for fluid control.

Liquid metal (LM) has emerged as a special matter and found a wide range of applications, including flexible electronics, soft robots, liquid marbles, nanomedicines, etc.[1–5], owing to its good mobility, high electrical conductivity. Among all liquid metals, Gallium (Ga) and Ga-based eutectic alloys stand out as one of the best candidates for commonly used liquid metals, mainly because of their excellent properties like nontoxicity and ease of preparation and transport[6,7]. In particular, dynamic behaviors of LM, such as transport and impact, are at the heart of practical applications. Thus, understanding and controlling the interfacial and transport properties of Ga-based LMs is of both fundamental and technical importance.

Different from conventional liquid droplets, whose mobility can be improved by passively engineering the underlying solid surface structures, tailoring the property of the liquid itself, or changing the ambient environments[8–14], enhancing the mobility of LM droplets is challenging. This is because, first, in ambient environments, the surfaces of Ga-based LMs, which possess large surface tension[15,16], are vulnerable to immediate oxidation. The formed thin oxide layer, mainly composed of $Ga_2O_3$ with a thickness of only a few nanometers[17], mechanically restrains the flow within LMs[18,19], and also leads to a strong adhesion of LM droplet to almost any solid surfaces[20,21]. Second, due to the inherent high density of the LM droplet, the gravity and inertia forces may become not negligible, which also affects the mobility of LM droplets on solid surfaces.

In this work, we discover a counterintuitive phenomenon that LM droplets can completely bounce off the water-covered solid surface with negligible adhesion, which is because the thin low-viscosity water layer acts as a lubrication layer that prevents LM droplets from contacting the solid substrate. Remarkably, we find that the mobility of LM droplets can be significantly enhanced by increasing the underlying

[1]Department of Mechanical Engineering, City University of Hong Kong, Hong Kong 999077, China. [2]Department of Mechanical and Aerospace Engineering, Southern University of Science and Technology, Shenzhen 518055, China. [3]School of Mechanical Engineering and Automation, Harbin Institute of Technology, Shenzhen 518055, China. [4]Department of Mechanical Engineering, Hong Kong Polytechnic University, Hong Kong 999077, China. [5]These authors contributed equally: Yuhang Dai, Minfei Li, Bingqiang Ji. ✉e-mail: steven.wang@cityu.edu.hk; haoc@hit.edu.cn; zk.wang@polyu.edu.hk

water layer thickness. This is in striking contrast to conventional observations that a highly viscous liquid layer suppresses the effective kinetic energy transformation owing to the dramatic extra viscous energy dissipation[8]. This intriguing phenomenon is not only fundamentally interesting, but also of great relevance in various practical applications, such as heat dissipation, simple flexible circuit fabrication, and 3D-printed robot for enhanced jumping ability and mobility[22–24].

## Results

### LM droplet bouncing experiments

We start our experiment by studying the dynamic interaction of LM droplets with a glass substrate coated with water layers of different thicknesses $h$. Specifically, a LM droplet with a volume of 14 µL (radius $R = 1.5$ mm) is released from a nozzle at a height $h_0$ of 40 mm, and impinges on the substrate at a Weber number $We = \rho_m u^2 R / \gamma_{ma} = 10.7$. Here, $u$ is the impact velocity, $\rho_m$ and $\gamma_{ma}$ are the density and surface tension of the LM, respectively (Table 1). The droplet dynamics are recorded by a high-speed camera (Photron, FASTCAM SA4) at a frame rate of 6000 frames per second. We focus on the LM droplet impact on a solid surface covered by a water microlayer, thus varying the layer thickness between 20–1000 µm. Figure 1a presents selected snapshots of a droplet impacting a bare glass substrate (i.e., $h = 0$ µm). As expected, the LM droplet spreads to a pancake shape immediately at ~4.2 ms after contact with the glass substrates, further retracts to a cone shape at ~22.2 ms, and sticks to the glass surface due to a large adhesion force between the LM oxide layer and the glass surface[20] (Supplementary Movie 1).

In striking contrast, Fig. 1b shows a different scenario that the LM droplet can completely bounce off the underlying substrate at ~22.2 ms in the presence of a water layer with $h = 89$ µm, and the final bouncing height of $h_r = 4.6$ mm (Supplementary Movie 1). More counterintuitively, by increasing $h$ to 298 µm, the LM droplet demonstrates even higher bouncing capability with $h_r = 5.1$ mm, which indicates less kinetic energy of the LM droplet is lost during impact (Supplementary Movie 1). LM droplet bouncing is also observed at a higher Weber number. It even happens when the water layer thickness is increased by one order of magnitude ($h = 10$ mm) (Supplementary Fig. 1), indicating this phenomenon is robust in a wide range of conditions. Such results are distinctive from our previous work showing that water droplets stick to the solid substrate due to the significant energy dissipation from the viscous liquid layer[8]. The unique water layer can dramatically modify the impacting dynamics of LM droplets.

### Lubrication effect of water layer

To get into more detail about the dynamic interaction between LM droplets and the underlying substrates, we use a high-speed camera coupled with an inverted microscope from the bottom view to visualize the contact line dynamics of LM droplets during the impact process. For a bare glass substrate without the coverage of a liquid layer (i.e., $h = 0$ µm), the triple-phase contact line instantly spreads in a circle shape at ~1.1 ms, and continues to expand without retraction afterward, as shown in Fig. 1d (Supplementary Movie 2). On the contrary, when impacting the glass substrate covered with water layers, the outline of the LM droplet expands during the spreading process and recoils at the

retraction process, as shown in Fig. 1e (Supplementary Movie 2). No LM residuals are observed on the glass slide after the rebound of the LM droplet (Fig. 1e and Supplementary Fig. 2). Using reflection interference contrast microscopy (RICM) coupled with a high-speed camera (see methods), we confirm that there exists an interference fringe underneath the LM droplet throughout the bouncing process (Fig. 1f, Supplementary Fig. 3 and Supplementary Movie 3), suggesting the presence of an intact water cushion. The minimum water layer thickness $h_m$ for LM droplet at $We = 10.7$ is calculated to be $\sim 5 R O h_f^{8/9} W e_f^{-10/9} = 0.37$ µm, where $Oh_f = \mu_w / (\rho_m \gamma_{mw} R)^{0.5}$ is the Ohnesorge number that relates the viscous force to the inertial and surface tension forces, and $We_f = \rho_m R u^2 / \gamma_{mw}$ is the Weber number comparing the inertial effect and the capillary effect in the water film[11,25,26]. Such a theoretical minimum water layer thickness is larger than the critical rupture thickness of a water film ($< 0.1$ µm)[27–29]. Thus, both experimental observation and theoretical analysis confirm the preserving of a continuous water lubrication film, which prevents the LM droplet from contacting the solid substrate during impact. Taken together, the enhanced mobility of LM droplets endowed by the water lubrication layer is different from the modification of liquid metal dynamics using electrochemical means[21], magnetic fields[30], environment vapor control[31], surface encapsulation[2,32], fuel feeding[33], textured structures[34], and in an underwater environment[35].

### Restitution coefficient

The effect of the water layer on LM droplet bouncing dynamics can be quantified by calculating the restitution coefficient $e$, defined as the ratio between the droplet velocities after and before impact on the water layer. Figure 2a presents the variation of $e$ as a function of the dimensionless water layer thickness $H = h/R$. Briefly, for a comparatively small water lubrication level ($H \lesssim H_c$), $e$ monotonically increases with the increase in $H$. As shown in Fig. 2a, the defined threshold $H_c$ is estimated to be 0.15 in our experiment. However, when $H$ is larger than $H_c$, $e$ reaches a plateau around 0.35, independent of the water layer thickness. Additionally, our supplementary experiments at a higher We show the same trend of $e$, and $e$ decreases with increasing We (Supplementary Fig. 4). The effect of water layer can also be reflected by measuring the contact time between LM droplets and the underlying substrate. As shown in Fig. 2b, when gradually increasing $H$, the contact time drops from 15.4 ms to 11.8 ms, corresponding to a 23% reduction (Fig. 2b).

These results allow us to hypothesize that the underlying water film serves as a lubrication layer between the impacting LM droplet and the solid substrate. To validate our hypothesis, we study the dynamic interaction of core-shell droplets with the solid surface with and without a water layer, in which the LM droplet serves as the core and water as the shell (Fig. 2c). In this way, the LM droplet is entirely encapsulated by the water shell without exposure to air, where the maximum water shell thickness at the bottom of the LM droplet before impact is measured as $h_s$. Thus, we can obtain an effective water layer thickness $H = (h + h_s)/R$ for the core-shell droplet impact. As shown in Fig. 2a, the restitution coefficient of the core-shell droplet coincides with that of the bare LM droplet, suggesting that the water shell around the LM droplet can also serve as a lubrication layer to facilitate the rebound of the LM droplet. For the core-shell droplet with

**Table 1 | Physical properties of the liquids used in the experiments**

| Liquids | $\rho(kg\,m^{-3})$ | $\mu(mPa\,s)$ | $\gamma_{ma}(mN\,m^{-1})$ | $\gamma_{wa}(mN\,m^{-1})$ | $\gamma_{mw}(mN\,m^{-1})$ |
|---|---|---|---|---|---|
| Liquid metal (LM) | 6250 | 1.99 | 686 ± 3.0 | N/A | N/A |
| DI water | 998 | 1.0 | N/A | 71.6 ± 1.0 | 511.0 ± 0.5 |
| 10 wt% glycerin-water solution | 1021 | 1.3 | N/A | 70.1 ± 0.5 | 515.7 ± 3.1 |
| 20 wt% glycerin-water solution | 1047 | 1.8 | N/A | 68.8 ± 0.2 | 534.1 ± 6.4 |
| 40 wt% glycerin-water solution | 1099 | 3.7 | N/A | 66.4 ± 0.6 | 539.1 ± 1.1 |

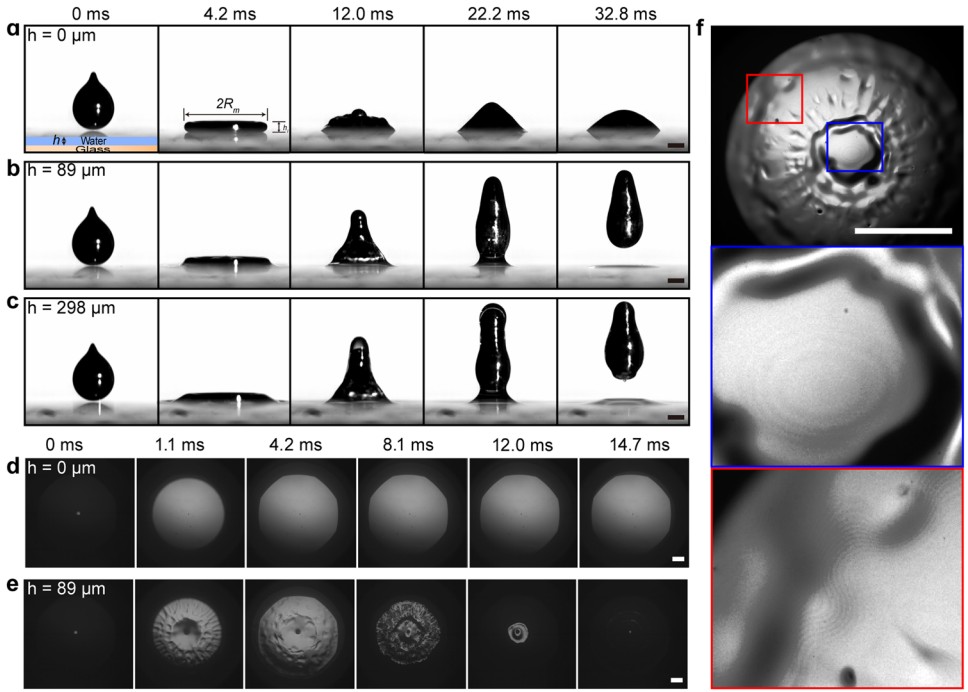

**Fig. 1 | Manifestation of LM droplet bouncing off a solid substrate covered with a water layer.** Selected snapshots of high-speed camera images showing the different impact patterns of LM droplets impinging on the surface with $h$ of 0 µm (**a**), 89 µm (**b**), and 298 µm (**c**). The selected bottom view of the snapshot captured by a high-speed camera by the inverted microscope with laser imaging system coupled with a high-speed camera shows the impact of LM droplets on a water layer-covered surface with (**d**) $h = 0$ µm and (**e**) $h = 89$ µm. The LM droplet radius $R = 1.5$ mm. **f** Interference fringe patterns imaged by the RICM confirm the existence of a continuous thin water layer during the impact process. The enlarged blue and red circled boxes clearly show the interference fringe patterns at the center and edge regions beneath the impacting LM droplet. The scale bars represent 1 mm.

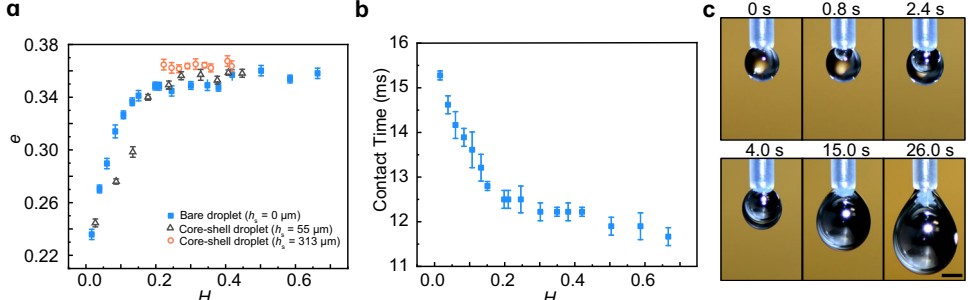

**Fig. 2 | Characterization of droplet bouncing dynamics. a** Restitution coefficient $e$ as a function of dimensionless water layer thickness $H$ for a bare LM droplet impact on a water layer-covered solid surface ($H = h/R$) and a water-coated LM droplet (core-shell droplet) impact on a clean solid surface ($H = (h + h_s)/R$). Here, $h$ is the water layer thickness at the glass slide, and $h_s$ is the thickness of the bottom shell of the core-shell droplet before impact. **b** Contact time as a function of dimensionless water layer thickness $H$ for a bare LM droplet impact on a water layer-covered solid surface. **c** Formation of core-shell droplets with LM serving as the core and water as the shell. The scale bar is 1 mm. The error bars for the contact time and restitution coefficient were obtained from the standard deviation of five replicate experiments.

$h_s = 313$ µm, $H \geq 0.21 > H_c$, indicating that the water shell layer around the LM droplet is thick enough to make $e$ reach the plateau region, and thus the restitution coefficient for the core-shell droplet remains a constant of about 0.35, consistent with that on a water layer of $H > H_c$ as shown in Fig. 2a.

## Theoretical model

We now propose a theoretical model to elucidate the counterintuitive behavior observed on the water layer. First, we consider the spreading of water on the impacting LM droplet by calculating the spreading coefficient of water on the LM surface, $S = \gamma_{ma} - \gamma_{mw} - \gamma_{wa}$. Given the condition of $S > 0$, the water spontaneously spreads on the LM droplet surface during impact, forming a water-coated LM surface (Fig. 3a). The time for the water film to spread along a distance of $R$ (droplet radius) can be estimated as $t_s \sim (\mu_m \rho_m R^4 / S^2)^{1/3} \sim O(1 \, \mathrm{ms})$ ($\mu_m$ is the LM viscosity)[36–38], which is of the same order of magnitude as the LM drop spreading time $t_s \sim (\rho_m R^3 / \gamma_{ma})^{1/2} \sim O(1 \, \mathrm{ms})$. Thus, the fast spreading of the water layer on the LM surface leads to the formation of a water meniscus with a radius of curvature $r$ connecting the faraway water layer at the glass substrate to the water-coated LM droplet surface, which also persists during the retraction stage as evidenced by our experimental images (Supplementary Fig. 5). Thus, the spreading droplet is subject to inertia, capillary force, and viscous force.

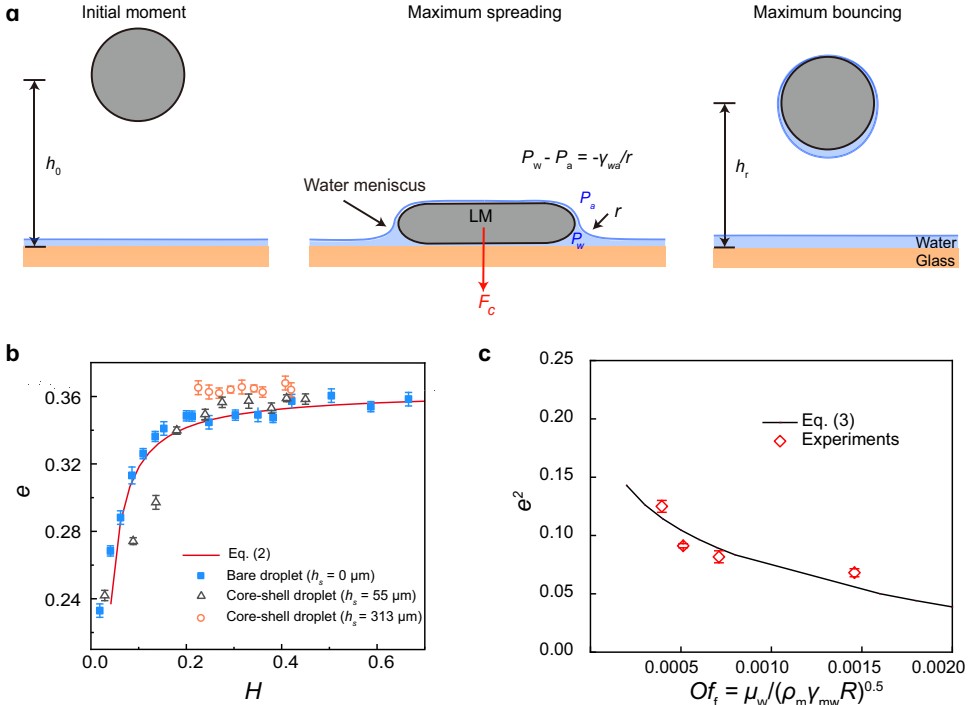

**Fig. 3 | Theoretical modeling of water layer-induced drop bouncing.**
**a** Schematic drawing showing that the LM droplet falls from a height of $h_0$ above the glass slide at the initial moment, and then deforms as a pancake shape and suffers from a capillary suction force $F_c$ during impact, and finally jumps to a maximum height of $h_r$. **b** Theoretical calculation of restitution coefficient $e$ as a

function of $H$ with Eq. (2), which is in good agreement with the experimental results. **c** Theoretical predictions of $e$ as a function of $Oh_f$ ($=\mu_w/(\rho_m\gamma_{mw}R)^{0.5}$) by Eq. (3) agrees well with the experimental results at $H > 0.2$. The error bars for restitution coefficient were obtained from the standard deviation of five replicate experiments.

In the initial moment of droplet falling, the total free energy of the droplet is written as $E_{p0} + E_{s0}$, where $E_{p0} = mgh_0$ and $E_{s0} = 4\pi\gamma_{ma}R^2$ are the gravitational energy and surface energy at the initial moment before falling, respectively. Upon in contact with the water layer-covered surface, the LM droplet is subject to a downward surface tension force $F_S \sim 2\pi R\gamma_{wa}$ at the intersection line between the water layer surface and the LM surface. Meanwhile, a negative Laplace pressure ($-\gamma_{wa}/r$) is developed across the water meniscus around the LM droplet (see Fig. 3a) and accordingly produces a downward capillary force $F_c \sim \pi R^2\gamma_{wa}/h$, in which we assume the radius of curvature of the meniscus $r \sim h$. Since $h \ll R$ for most experimental cases, $F_s$ is negligible compared to $F_c$. Considering that the acting distance of $F_c$ scales as $R$, the work done by the capillary force $F_c$ is written as $W_{Fc} = C_1\pi R^3\gamma_{wa}/h$, where $C_1$ is a fitting parameter that is associated with the radius of curvature of the water meniscus and the time variation of acting distance of $F_c$ during impact. Meanwhile, the energy of the LM droplet is also dissipated by viscous effect during impact, both within the LM droplet and the underlying liquid layer. The viscous dissipation inside the LM droplet is calculated to be $E_{\mu m} = 0.5E_{p0}$ where the continuous water layer is treated as a free-slip boundary condition[39]. The viscous energy dissipation in the underlying water layer can be estimated as $E_{\mu w} = C_2 h_w R^2\mu_w(u/h_w)^2\tau$ based on the lubrication approximation theory[8,39], where $h_w \sim 5ROh_f^{8/9}We_f^{-10/9}$ is the water layer thickness scaled using the minimum water layer thickness upon impact, $\tau \sim (\rho_m R^3/\gamma_{ma})^{1/2}$ is the contact time, and $C_2$ is a fitting parameter considering the time variations of the droplet shape and the velocity field in the liquid layer during impact. At the final moment when the droplet bounces to the maximum height, the free energy of the LM droplet is expressed as $E_p + E_s$, where $E_p = mge^2 h_0$ and $E_s \approx E_{s0}$ are the gravitational energy and surface

energy, respectively. Thus, according to the energy conservation, we have

$$E_{p0} + E_{s0} - W_{Fc} - E_{\mu m} - E_{\mu w} = E_p + E_s, \qquad (1)$$

from which the droplet restitution coefficient $e$ is obtained as

$$e = \left\{ \frac{1}{2} - C_1 \cdot \frac{3}{2We} \cdot \frac{\gamma_{wa}}{\gamma_{ma}} \cdot \frac{1}{H} - C_2 \cdot \frac{3}{10\pi} \cdot We^{10/9} Oh_f^{1/9} \left( \frac{\gamma_{mw}}{\gamma_{ma}} \right)^{-11/18} \right\}^{0.5} \qquad (2)$$

Figure 3b shows the variation of the restitution coefficient $e$ predicted based on Eq. (2) as a function of $H$, in which $C_1$ is 0.2 and $C_2$ is 0.535. Notably, Eq. (2) quantitatively agrees with our experimental results with the water layer. When $h$ is comparable to $R$ ($H > 0.2$), $F_c$ and $W_{Fc}$ become negligible. As a result, the restitution coefficient becomes independent of the water layer thickness. Furthermore, we conducted additional experiments using a surfactant aqueous liquid layer containing sodium dodecyl benzene sulfonate (Supplementary Fig. 6), which exhibits lower interfacial tensions compared to pure water. We found the dependence of the restitution coefficient $e$, on the dimensionless water layer thickness $H$, for the surfactant case is similar to that of the pure water layer case. Remarkably, the predictions from Eq. (2) align well with the experimental results for both cases, further confirming the validity of our theoretical model.

To probe the effect of the liquid layer viscosity on the restitution coefficient of a bouncing droplet, we choose the glycerin solutions with different viscosities but similar interfacial tensions (Table 1), which can be achieved by tailoring the glycerin mass fraction ranging from 10% to 40% (Supplementary Fig. 7). Note that in our experiment, we intentionally select a large liquid layer thickness (at the plateau

region of $e$) to decouple its effect on $e$, while $We$ is fixed at 10.7. Because $W_{Fc}$ is negligible at a large $H$, Eq. (2) can be further reduced to

$$e = \begin{cases} (0.5 - 0.92Oh_f^{1/9})^{0.5}, & \text{for } Oh_f \leq 0.004 \\ 0, & \text{for } Oh_f > 0.004 \end{cases} \quad (3)$$

As shown in Fig. 3c, the data predicted based on Eq. (3) demonstrate excellent agreement with our experimental findings, despite the variations in the viscosity of the liquid layer. When $Oh_f > 0.004$, the LM droplet cannot bounce off due to the large viscous dissipation in the liquid layer, yielding $e = 0$, which is validated by our experiments with a highly viscous aqueous liquid layer with $Oh_f = 0.0043$ (Supplementary Fig. 8).

## Discussion

The complete bouncing of the LM droplet observed in this study differs from our previous work on water droplet impact on slippery oil-infused surfaces[8]. In the previous study, a highly viscous oil layer (viscosity ≥ 0.15 Pa s) was deliberately chosen, resulting in $Oh_f \geq O$ (0.1). However, in the present study, the aqueous liquid layer above the solid surface has a lower viscosity, leading to $Oh_f \leq O$ (10⁻³). Consequently, the viscous dissipation is significantly reduced, facilitating the bouncing of the droplets. The counterintuitive bouncing dynamics of the LM droplet originate from the trapping of a thin low-viscosity lubrication film that prevents droplet-solid contact with minor viscous dissipation, triggering the complete rebound of the LM droplet. Moreover, the negative capillary pressure in the thin water lubrication film due to the spontaneous spreading of water on the LM droplet modulates the LM droplet's restitution coefficient.

The intriguing phenomenon reported here has potential applications in a wide range of scenarios[40,41]. By leveraging the jumping feature of LM droplets, we present the proof-of-concept demonstration of a water strider-like robot that integrates LM droplets on its legs to serve as reconfigurable wheels. In our design, LM droplets with a volume of ~14 μL are attached to the bottom of four legs of a 3D-printed water strider robot due to the strong adhesion force generated by liquid metal oxide (Fig. 4a). As a demonstration, the as-fabricated water strider-inspired robot is released from a height of 40 mm. When in contact with the water layer, the thin water lubrication layer prevents direct contact between the robot wheels and the solid surface, enabling the water strider robot to successfully jump from the sticky water layer. Moreover, spontaneous sliding locomotion along the surface with negligible adhesion is achieved, by placing it on a 30° inclined substrate, as shown in Fig. 4b. Notably, the measured jumping

height and sliding velocity are 2.61 mm and 0.34 mm s⁻¹, which are 3.5 times and 1.5 times higher than those of counterparts without the decoration of LM droplets, respectively (Fig. 4c). Therefore, such an enhanced jumping and sliding capability switched by the thin water layer adds a new dimension and flexibility for robot locomotion.

In general, we report a peculiar phenomenon that the bouncing performance of LM droplets can be regulated by the thickness of the underlying water layer. We highlighted that the bouncing of the LM droplet is facilitated by the existence of the low-viscosity aqueous layer above the glass slide that prevents the LM droplet from contacting the solid surface with low viscous dissipation, which is strikingly different from our previous observation that a viscous oil layer completely inhibits the bouncing due to the significant viscous energy dissipation in the oil layer. In particular, the restitution coefficient increases with water layer thickness before reaching a plateau. Our theoretical analysis points out that the liquid layer thickness modulates the negative capillary force within the aqueous liquid layer and thus changes the restitution coefficient. The largely unexplored bouncing regime and the mechanisms we revealed advance our fundamental understanding and on-demand manipulation of liquid behaviors for practical applications.

## Methods

### Materials

Gallium-based alloy, named eutectic gallium indium, or EGaIn (75% Ga, 25% In, Sigma Aldrich), was used as the liquid metal (LM). The liquid layer used in the experiments is deionized (DI) water, or the mixture of glycerin (AR, 99%, Macklin) and DI water with different glycerin mass fractions (10–40%). The surface tensions and interfacial tensions of these liquids were measured using the pendant drop method, and the results are listed in Table 1, along with their density $\rho$ and viscosity $\mu$.

### Preparation of water layer on the glass surface

The glass substrates were treated with oxygen plasma to remove the dust and render it superhydrophilic before using for experiments. The liquid layer on the glass surface was formed by directly dripping the DI water or glycerin solution droplet to conduct a fast spontaneous spreading, and the thickness of the liquid layer was controlled by the total volume of water droplets and measured using the probe method.

### Water lubrication effect experiments

The LM droplets were released from a polyethylene tube with an inner diameter of 0.38 mm connected to a 1 mL syringe. A flow

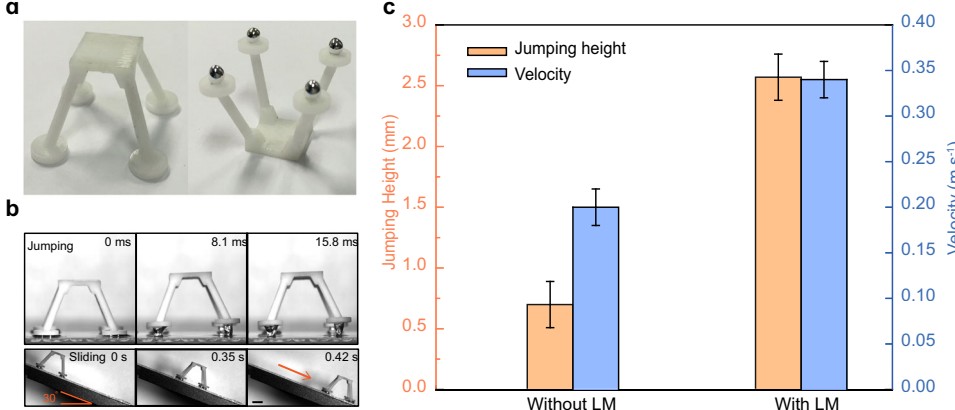

**Fig. 4 | Extending water layer-induced drop bouncing for robot application.** **a** Construction of 3D printed robot equipped with four LM droplets at its feet. **b** Optical images show the jumping and rapid sliding of the robot equipped with LM droplets. **c** Comparison of jumping height and sliding velocity for robots decorated with and without LM droplets. The scale bar is 10 mm. The error bars for the jumping height and velocity were obtained from the standard deviation of five replicate experiments.

rate of 0.06 sccm was controlled by a syringe pump. The exposure time of the LM droplet in the air before impact was about 15 s for all the experimental cases, which ensured that the LM droplet surface was oxidized[20,42,43]. Using such setup, LM droplet of a radius of about 1.5 mm (the volume of 14.1 μL) was generated and released from a pre-determined height of $h_0$. The LM droplet dynamics were recorded by the high-speed camera (Photron, Fastcam Nova S) with a frame rate of 6000 fps, and the inverted microscope (Nikon Ti-s) with a laser imaging system coupled with a high-speed camera was used to capture the bottom view of the droplet impact process. The experiments were conducted at room temperature (22 ± 2 °C) and ambient environment (1 atm).

## Water layer thickness measurement

We used the probe method to measure the thickness of the water layer. On two mobile microscope platforms (QuantaTM 450; FEG), the superhydrophilic glass slide and the hydrophilic hanging probe were fixed horizontally and vertically, respectively. We first recorded the vertical position of the hanging probe, of which the tip just touched the glass slide surface. Then, we removed the probe vertically and wet the superhydrophilic glass slide with a water droplet to form a water layer with the required thickness. We recorded the vertical position of the hanging probe again after the hanging probe slowly moved down until its tip touched the water layer surface, forming a meniscus on the tip[44,45]. We could get the exact water layer thickness by such two position records. The measurement error is 1 μm.

## Surface tension measurement

Surface tensions of the LM ($\gamma_{ma}$) and liquid layers ($\gamma_{wa}$), and the interfacial tension ($\gamma_{mw}$) of LM droplets exposed to the water or glycerin solution (used as the liquid layer) were measured by the pendant drop method with a Contact Angle Meter (TBU 100, Dataphysics)[15,17,46–49]. LM or aqueous liquid droplets at room temperature were generated at a volume rate of 6 μL/min on the pinpoint to form a pendant drop. The profile of the pendant droplet at the equilibrium state was captured by the contact angle meter. Our measurements give a LM surface tension of $\gamma_{ma} \approx 686 \text{mNm}^{-1}$, and a LM-water interfacial tension of $\gamma_{mw} \approx 511 \text{mNm}^{-1}$, which are consistent with the previous reports[15,23,24].

## Water layer visualization using the interference fringe

The interference images were captured using RICM combined with a high-speed camera (Photron, Fastcam Nova S), at a frame rate of 8000 fps and a shutter speed of 1/100,000. A high-pressure mercury lamp with a main wavelength range of 405–546 nm is used as the light source.

# Data availability

The experimental raw data supporting the findings of this study are available from the corresponding author upon request. The processed data are available within the manuscript and its Supplementary Information Files. Source data are provided with this paper.

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

## Acknowledgements

We acknowledge support from the Research Grants Council of Hong Kong (no. 1-BDCN, no. C1006-20WF and no. B-QC0R), the Tencent Foundation through the XPLORER PRIZE, the Innovation and Technology Council (no. 9440248), the National Natural Science Foundation of China (grant nos. 51975502, 21621001, 12071367, 52005128, and 91852205), and Program of Shenzhen Peacock Innovation Team (no. KQTD20210811090146075).

## Author contributions

Z.W. conceived the research. S.W., C.H., and Z.W. supervised the research. Y.D. and M.L. carried out the experiment. Y.D., B.J., X.W., and S.Y. collected data and analyzed the experimental results. B.J. and P.Y. conducted the theoretical modeling. M.L., Y.D., and S.Y. prepared the experimental apparatus. All authors contributed to the interpretation and drafting of the paper.

## Competing interests

The authors declare no competing financial or non-financial interests.
