## [Peer Review File · Nature Communications]

Liquid metal droplets bouncing higher on thicker water layerREVIEWER COMMENTS

Reviewer #1 (Remarks to the Author):

The phenomenon reported in this manuscript is very interesting. To decrease the adhesion of liquid metal droplets on substrates and thus increasing its mobility is an interesting topic. To manipulate the oxidation layer of the eutectic liquid metal alloy is also an important task in the field. However, the explanation provided in this manuscript is not convincing. More experimental results need to be analyzed so as to better understand the phenomenon, and more experimental details need to be provided.

These major and minor issues are listed below:

Major issues:

During impact, the contact time between the liquid metal droplet and the water film is less than 15 ms. That is to say, the old oxidation film on the liquid metal droplet which is formed in air should be oxidized again by water in less than ~ 5 ms to facilitate droplet bouncing. However, as far as I know, there is no proof that such an oxidation process can happen in such a short time scale, because it is difficult to quantify the time required to form the oxidation film, either in air or in water.

So, the evidence provided in the manuscript is not enough to prove that this phenomenon is caused by a sudden increase in the surface tension of the liquid metal droplet. While explanation in this manuscript is unconvincing, I find another explanation which might be more plausible.

In this explanation, the liquid metal droplet can bounce off because the water film on the substrate prevents the liquid metal from contacting the solid substrate. During the impact, the oxide-water interface becomes rough, see figure 1e. There is an upper bound for this roughness, thus there is a critical water film thickness above which the liquid metal droplet does not touch the substrate at all. This corresponds to $h^* > h_c$ where the restitution coefficient is more or less a constant. When the water film thickness increases from 0 to this critical value, the contact area between the liquid metal droplet and the substrate decreases continuously, thus the restitution coefficient increases continuously, this corresponds to the part where $h^* < h_c$. For the core-shell droplets, it is possible that the water shell on the droplet is already thick enough so that the liquid metal cannot touch the substrate at all. This leads to another question, what is the thickness of the water shell of the core-shell droplets? This shell thickness should be added to the water film thickness on the substrate to calculate h^* . Plus, if this

combined water film thickness is above the critical value, then it is no wonder that the restitution coefficient reaches the plateau directly.

Minor issues:

1. As mentioned above, for the core-shell droplets, what is the thickness of the water shell?
2. How long are the liquid metal droplets exposed in air before they are released? Is this time kept constant or long enough to ensure that the oxidation layer is the same between different tests?
3. How exactly is h_{LM} determined? Even at maximum spreading of the droplet, the top surface of the droplet is not necessarily flat, so is h_{LM} measured at the highest point of the pancake-shaped droplet? Why not indicate the definition of h_{LM} in figure 1, for example? Then, how does this measured h_{LM} vary between different tests? I.e., is this quantity reproducible?
4. According to Ref. 32, the spreading parameter of water on Ga_2O_3 is $S = 300 - 500 - 72 \text{ mN/m} < 0$. This means that water will not spread completely on $EGaIn$, but meet the liquid metal with a contact angle. Then why water can spread to the top of the liquid metal droplet in less than 15 ms during the impact? Plus, supplementary figure 1 is too vague to see that water is indeed on top of the liquid metal droplet. The contrast of the image needs to be increased. For example, why not use fluorescent dye and illuminate the droplet with a laser beam?
5. How do the authors get the error bar in figure 2, 3, 4, 5? Are the corresponding experiments repeated? If so, how many times?
6. When $h^* < h_c$, equation (4) is a square root function of h^* , then why in figure 4b the theory predicts a straight line for $h^* < h_c$? Plus, though the theoretical prediction seems to agree with the experimental results in figure 4b, the theoretical model which assumes the droplet at maximum spreading as a column is too far from reality.

In conclusion, a much more convincing explanation needs to be provided, and all the minor issues need to be addressed before the manuscript is publishable.

Reviewer #2 (Remarks to the Author):

The article, "Water sculpting effect:.....liquid film" demonstrates- according to the authors, a striking new phenomenon. However, the non-sticky nature of the oxide in the presence of the water/slip layer has been demonstrated several times in the literature. The current article, therefore, has insignificant novelties to be considered for Nat. Comm.

Literature survey: There are several articles published by Dickey Group (NCSU), Dickey in collaboration with Majidi (CMU), Shiyang Tang (UK), Khoursh (Australia), and Jing Liu (China), where the fluid-liquid metal interactions were explained in several ways. The influence of water on Gallium alloys was also dissected utilizing the contact angle and residual adhesion in one paper published in ACS AMI. The jumping liquid metal droplet work published by Dr. Dickey has phenomena similar to the current article. In fact, an oxidized metal droplet is hydrophilic. In the presence of water, the surface oxide changes. The low tension of the new oxide allows water to wet the adjacent surface. Therefore, it is quite intuitive to see the sculpting effect. Overall, this work seems fit elsewhere.

Reviewer #3 (Remarks to the Author):

In this paper the authors consider liquid metal droplets that rebound (jump) upon impact on a prewetted surface. The authors provide experimental results and propose a physical mechanism leading to the observed phenomenon. The mechanism is based essentially on a modification of surface tension of the impacting metal; such modification depends on the thickness of the prewetted layer, leading to larger jumps for the thicker ones.

The results are quite interesting, and as the authors point out, may be relevant not only as a peculiar fluid mechanical phenomena, but also in applications. One of these (walking robot) is outlined in the manuscript. So in principle I could be encouraging to move ahead with the reviewing process, however, the authors should improve the manuscript and their results first. Some rather crucial points are confusing at this point and these need to be clarified not to confuse the reader. I'll list my comments in a random order below - in my opinion, all of these should be addressed carefully before manuscript could be considered for publication.

The presentation: this needs some work - sculpting effect (where is this coming from?); page 2 - 'magic knife' - I don't think there is any magic here, please use less flowery language; 'sculpts the sticky skin' - well now, in addition to 'sculpts' we need to explain 'sticky' and 'skin'. I think I know what the authors mean, but this type of wording (and lack of justification) are not appropriate.

The coefficient of restitution is -not- proportional to the water thickness as specified in the abstract - please be more precise.

The length scale - why using h_{LM} - thickness at max drop size - as the length scale? This one is different for each experimental condition, and I'm not sure the corresponding values are specified. Therefore the reader cannot bring the results back to the dimensional form. This is a serious problem.

What happens for even thicker water films? Do drops still rebound? What happens in the limit where water film thickness is orders of magnitude larger?

Page 5 says that experimental observation for the critical water film thickness is 226.2nm. I do not believe that the authors are able to measure this quantity so precisely. What are the error bars?

The fitting parameter alpha on page 5 comes as a surprise. Why is it needed and where is it coming from?

References. The author do not cite the papers considering jumping metal drops in the context where such jumping is caused by the initial conditions. While such context is different from the present one, it is relevant, since it provides a useful example of another setup where jumping droplets are possible. See Science 309, 2043 (2005) and PRL 111, 034501 (2013).

Point-to-point Responses to Referees' Comments

Referee #1 (Remarks to the Author):

The phenomenon reported in this manuscript is very interesting. To decrease the adhesion of liquid metal droplets on substrates and thus increasing its mobility is an interesting topic. To manipulate the oxidation layer of the eutectic liquid metal alloy is also an important task in the field. However, the explanation provided in this manuscript is not convincing. More experimental results need to be analyzed so as to better understand the phenomenon, and more experimental details need to be provided.

Response: We thank the referee for the time spent on our paper, constructive comments, and remarks. We are pleased that the referee found our work "very interesting." Empowered by the referee's concern about the theoretical explanation, we have revised our previous model, conducted detailed theoretical analyses, and proposed a new theoretical model which reasonably explains our experimental observations. Besides, we have also conducted additional experiments and provided convincing experimental details. We have revised the paper according to the referee's comments and believe that our paper's scientific merit is much improved. The details of the changes that we made are shown in the following responses, and changes to the text are highlighted in the text body.

Comment 1: During impact, the contact time between the liquid metal droplet and the water film is less than 15 ms. That is to say, the old oxidation film on the liquid metal droplet which is formed in air should be oxidized again by water in less than ~ 5 ms to facilitate droplet bouncing. However, as far as I know, there is no proof that such an oxidation process can happen in such a short time scale, because it is difficult to quantify the time required to form the oxidation film, either in air or in water.

So, the evidence provided in the manuscript is not enough to prove that this phenomenon is caused by a sudden increase in the surface tension of the liquid metal droplet. While explanation in this manuscript is unconvincing, I find another explanation which might be more plausible.

In this explanation, the liquid metal droplet can bounce off because the water film on the substrate prevents the liquid metal from contacting the solid substrate. During the impact, the oxide-water interface becomes rough, see Figure 1e. There is an upper bound for this roughness, thus there is a critical water film thickness above which the liquid metal droplet does not touch the substrate at all. This corresponds to $h^* > h_c$ where the restitution coefficient is more or less a constant. When the water film thickness increases from 0 to this critical value, the contact area between the liquid metal droplet and the substrate decreases continuously, thus the restitution coefficient increases continuously, this corresponds to the part where $h^* < h_c$. For the core-shell droplets, it is possible that the water shell on the droplet is already thick enough so that the liquid metal cannot touch the substrate at all. This leads to another question, what is the thickness of the water shell of the core-shell droplets? This shell thickness should be added to the water film thickness on the substrate to calculate h^* . Plus, if this combined water film thickness is above the critical value, then it is no wonder that the restitution coefficient reaches the plateau directly.

Response: We are lucky to have the referee who provided very insightful and constructive comments and suggestions that pointed out a new direction and fundamentally improved the scientific depth of our work. In particular, the referee's comment on the presence of water film prevents contact between the impacting liquid metal and underlying substrate inspired us to derive a vigorous model to fully substantiate our findings.

First, we experimentally and theoretically demonstrate that the water layer indeed prevents contact between the LM droplet and the glass substrate. Experimentally, the bottom view visualization shows that the bouncing of LM droplets can only occur when a thin water film exists (Figure 1e). Moreover, no LM residuals are observed on the glass slide after the rebound of the LM droplet, indicating that the liquid film acts as a lubrication layer preventing the LM droplet from contacting the solid substrate. In contrast, without the presence of water film, the contact line is pinned at the glass slide (Figure 1d).

The existence of a continuous water lubrication layer during impact is also consistent with our theoretical analysis. The minimum fluid film thickness between the impacting drop and the solid substrate is estimated as $h_m = 5ROh_f^{8/9}We_f^{-10/9}$, where $Oh_f = \mu_w/(\rho_m\gamma_{mw}R)^{0.5}$ and $We_f = \rho_m R u^2/\gamma_{mw}^{1-3}$. In this study, the minimum water layer thickness for LM droplet with $We = 10.7$

is calculated to be $\sim 0.37 \mu\text{m}$, larger than the critical rupture thickness of a water film ($< 0.1 \mu\text{m}$)^{4,5}, indicating that a continuous and stable water lubrication layer can be preserved during impact.

Second, to establish a theoretical model to interpret the evolution of the droplet restitution coefficient e , we first consider the spreading coefficient of water on LM droplet, expressed as $S = \gamma_{ma} - \gamma_{mw} - \gamma_{wa}$. In our study, the spreading coefficient is 103mN/m , suggesting that the water can spontaneously spread on the LM droplet surface, forming a water-coated LM surface. Moreover, from the timescale perspective, the time for the water film to engulf the whole droplet is estimated as $t_s \sim (\mu_m \rho_m R^4 / S^2)^{1/3} \sim 0(1 \text{ ms})$ ⁶⁻⁸, which is one order of magnitude smaller than the droplet contact time (Figure 2b). Hence, the LM droplet is fully engulfed by a thin water film (Figure 4a, also see Figure R1 here).

Figure R1. Schematic drawing showing drop impact process at the maximum spreading diameter and the maximum jumping height.

We then analyze the forces acting on the impacting LM droplet. The spreading of water forms a meniscus with a radius of curvature r connecting the faraway water layer at the glass substrate to the water-coated LM droplet surface. This meniscus exerts a downward surface tension force F_s ($\sim 2\pi R\gamma_{wa}$) at the intersection line between the water layer surface and the water-coated LM surface, and also a downward capillary force F_c produced by the negative Laplace (or capillary) pressure ($P_w - P_a = -\gamma_{wa}/r$) across the meniscus. Assuming $r \sim h$, we have $F_c \sim \pi R^2 \gamma_{wa} / h$. The height difference of the spreading LM droplet between two critical time points,

i.e., immediately before bouncing off the surface and reaching the maximum spreading, can be scaled as R . The forces from the gas can be neglected considering the small density and viscosity ratios of gas to liquids. The viscous effects are also negligible due to the small Ohnesorge numbers of water and LM, i.e., $Oh_w = \mu_w/(\rho_w \gamma_{wa} R)^{0.5} = 0.0027 \ll 1$ and $Oh_m = \mu_m/(\rho_m \gamma_{ma} R)^{0.5} = 0.0007 \ll 1$. Correspondingly, we neglect the inertia and gravity effects from water due to its small volume and density compared to those of LM. Considering the energy conservation of the LM droplet, we have

$$2\pi(R_m h_l + R_m^2)\gamma_{mw} - 2\pi a R^2 \gamma_{wa} - \pi b R^3 \gamma_{wa}/h = mgh_r + 4\pi R^2(\gamma_{mw} + \gamma_{wa}) \quad (R1)$$

The left part of the equation represents the surface energy of the LM droplet at the maximum spreading and the work done by F_s and F_c , and the right part of the equation refers to the gravitational potential energy and the surface energy of the LM droplet at the maximum bouncing height, with $h_r = e^2 h_0 = 0.5e^2 u^2/g$. Here, a and b are fitting parameters considering the variation of the drop shape and lift distance and the meniscus inclination angle during retraction. h_l is the droplet thickness at maximum spreading and equals $4R^3/(3R_m^2)$, considering the droplet deforms as a disk (Figure 1). Deduced from Eq. (R1), the droplet restitution coefficient e can be written as

$$e = \left\{ \frac{3}{We} \cdot \frac{\gamma_{mw}}{\gamma_{ma}} \cdot \left(\frac{4}{3} R_m^{*-1} + R_m^{*2} \right) - \frac{6}{We} \cdot \frac{\gamma_e}{\gamma_{ma}} - \frac{3a}{We} \cdot \frac{\gamma_{wa}}{\gamma_{ma}} - \frac{3b}{2We} \cdot \frac{\gamma_{wa}}{\gamma_{ma}} \cdot \frac{1}{H} \right\}^{0.5} \quad (R2)$$

where $R_m^* = R_m/R$, and $\gamma_e = \gamma_{mw} + \gamma_{wa}$ is the effective surface tension of the water-coated LM droplet after bouncing off. Eq. (R2) indicates that the droplet restitution coefficient e decreases with the decreasing dimensionless water layer thickness H . This is because a thinner water lubrication layer exerts a larger Laplace pressure as well as a larger capillary suction force F_c , thus resulting in a smaller bouncing velocity and a smaller e , which well explains our experimental observations. By adopting $a = 2$ and $b = 0.2$ and using the experimental measured $R_m^* = 2.16$, Eq. (R2) quantitatively agrees with our experimental results (Figure 3b).

Finally, this theoretical analysis is also consistent with the interpretation for the core-shell droplets impact proposed by the referee. The thickness of the water shell at the bottom of the LM droplet is measured to be about 313 μm , which is larger than the critical water layer thickness (226 μm), indicating the water layer beneath the impacting LM droplet is thick enough to make e reach the plateau region. This explains why the restitution coefficient of the

core-shell droplets remains a constant of about 0.35 at various water layer thicknesses (see Figure 2a), as suggested by the referee.

In conclusion, we reveal the complete rebound of LM droplets originates from the trapping of a thin water film underneath that prevents droplet-solid contact, and the restitution coefficient is modulated by the negative capillary pressure as a result of the spontaneous spreading of water on LM droplets.

In order to clarify the above points, we have revised the manuscript by adding explanations in Lines 88-100 and Lines 123-177.

Comment 2: As mentioned above, for the core-shell droplets, what is the thickness of the water shell?

Response: As discussed in our response to Comment 1, the thickness of the water shell at the bottom of the LM droplet is measured to be about 313 μm . To provide more detailed information and a deeper understanding, we have provided the water shell thickness at Lines 117-120 in the revised manuscript.

Figure R2. Snapshot of the core-shell droplet before contact. A water shell layer is clearly observed around the LM droplet. The water shell thickness at the bottom of the LM droplet is measured to be 313 μm . The scale bar is 1mm.

Comment 3: How long are the liquid metal droplets exposed in air before they are released? Is this time kept constant or long enough to ensure that the oxidation layer is the same between different tests?

Response: We thank the referee for this helpful comment. In our experiment, we use a syringe pump (Model 200, kd Scientific) with a constant flow rate (60 $\mu\text{L}/\text{min}$) to generate the liquid metal droplet from a needle of constant size. Thus the exposure time of the liquid metal droplets to the air is about 15 s and keeps consistent between different tests. According to previous studies (Torben Daeneke, Science. 2017) ⁹⁻¹¹, the oxidation layer at the LM surface is about $O(1 \text{ nm})$, and the oxidation of the fresh LM surface is very fast with a reaction time $\ll 1 \text{ s}$. Thus, the exposure time of the LM droplet in our experiments is long enough to ensure that the oxidation layer is the same between different tests.

We have added more explanation in Methods at Lines 213-215 to clarify the above points.

Comment 4: How exactly is h_{LM} determined? Even at maximum spreading of the droplet, the top surface of the droplet is not necessarily flat, so is h_{LM} measured at the highest point of the pancake-shaped droplet? Why not indicate the definition of h_{LM} in Figure 1, for example? Then, how does this measured h_{LM} vary between different tests? I.e., is this quantity reproducible?

Response: We thank the referee for this comment. In our original manuscript, we define h_{LM} as the maximum thickness of the droplet at the maximum spreading. h_{LM} is measured at the highest point of the droplet. h_{LM} remains the same at a fixed We even though the water layer thickness varies, as shown in Figure R2. Each experiment is conducted five times, and the data points in the figure represent the average of five experiments, indicating our experimental measurements are reproducible.

In the revised manuscript, we use the LM droplet radius R as a characteristic length scale to non-dimensionalize the water layer thickness h . h_{LM} (h_l in the revised manuscript) is only used in Eq. (1) and estimated as $4R^3/(3R_m^2)$ according to volume conservation based on a disk shape assumption at maximum spreading. Since the measured maximum drop spreading diameter is R_m and is independent of We , this estimation does not change the conclusion that h_{LM} remains the same at the same We .

In order to clarify the above point, we have added the definition of h_{LM} (h_l in the revised manuscript) in Figure 1a.

Figure R3. LM droplet at maximum spreading (a) Snapshots of $We=10.7$, scale bar represents 1 mm. (b) h_{LM} for LM droplet under the same We with different water layer thickness h .

Comment 5: According to Ref. 32, the spreading parameter of water on Ga_2O_3 is $S = 300 - 500 - 72 \text{ mN/m} < 0$. This means that water will not spread completely on EGaIn, but meet the liquid metal with a contact angle. Then why water can spread to the top of the liquid metal droplet in less than 15 ms during the impact? Plus, supplementary figure 1 is too vague to see that water is indeed on top of the liquid metal droplet. The contrast of the image needs to be increased. For example, why not use fluorescent dye and illuminate the droplet with a laser beam?

Response: We thank the referee for this insightful comment. We noted that our measured value of 0.3 N/m based on the maximum height method is not consistent with the literatures. We thus re-measured the surface tension of LM γ_{ma} and the interfacial tension of LM-water interface γ_{mw} using the pendant drop method, a method more widely used for surface tension measurement (see Methods section in the revised manuscript). The re-measured $\gamma_{ma} = 0.686 \text{ N/m}$, which is consistent with previous literatures¹²⁻¹⁶. While the measured $\gamma_{mw} = 0.511 \text{ N/m}$, which agrees with our previous measurement using the maximum height method and also the reported values^{14,17}.

As mentioned in our response to Comment 1, the spreading coefficient ($S = \gamma_{ma} - \gamma_{mw} - \gamma_{wa} = 103 \text{ mN/m} > 0$) of water on LM surface is positive, meaning that the water spontaneously spreads on the LM droplet surface during impact. The time for the water film to engulf the whole droplet can be estimated as $t_s \sim (\mu_m \rho_m R^4 / S^2)^{1/3} \sim O(1 \text{ ms})$ based on previous studies⁶⁻⁸, which is one order of magnitude smaller than the drop contact time (see Figure 2b), indicating that water can spread to the top of the liquid metal droplet (fully engulfed) in less

than 15 ms during the impact.

We indeed observed the fully engulfing of the water film on the LM droplet using the current experimental setup, as shown in Figure R4 (see also Supplementary Figure S3). A thin water shell can be clearly observed after the drop retraction, and it also exists in the jet drop generated by the drop impact (Figure R4a). It is a pity that we cannot conduct extra experiments using a laser beam to increase the contrast for clear imaging of water engulfing owing to the limitation of our lab equipment and the inconvenience during the pandemic. But the comment raised by the referee points out an important direction for our future work.

In order to clearly identify the above points, we have provided the re-measured surface tensions at Line 64-66, introduced the pendant drop method for surface tension measurement at the Methods section, given theoretical explanations and experimental evidence on the water spreading and LM droplet fully engulfing at Line 127-133 and Supplementary Figure S3.

Figure R4. Experimental images showing that the LM droplet is fully engulfed by a thin water film after bouncing off the water layer-covered glass slides. (a) $We = 18.76$, $H = 0.25$. (b) $We = 18.76$, $H = 0.16$. The droplet radius $R = 1.5$ mm. The scale bar is 1mm.

Comment 6: How do the authors get the error bar in figure 2, 3, 4, 5? Are the corresponding experiments repeated? If so, how many times?

Response: The error bars in Figures 2-5 are calculated as the standard deviation of five replicate experiments.

Comment 7: When $h^* < h_c$, equation (4) is a square root function of h^* , then why in figure 4b the theory predicts a straight line for $h^* > h_c$? Plus, though the theoretical prediction seems to agree with the experimental results in figure 4b, the theoretical model which assumes the droplet at maximum spreading as a column is too far from reality.

Response: We thank the referee for this insightful comment. In our original manuscript, we drew the straight line for $h^* > h_c$ using the e value at $h = h_c$ predicted by the theoretical model, based on the experimental observation that e remains constant at $h^* > h_c$. In our revised manuscript, we propose a new theoretical model following the referee's suggestion, which predicts the experimental results well and captures the main underlying physics at the same time. In our revised manuscript, the denotation of characteristic thickness h^* is replaced by H . In the revised Figure 3(b), the solid line represents the theoretical prediction at both $H < h_c$ and $H > h_c$.

We agree that it is elusive to quantitatively describe the shape of a droplet at the maximum spreading. However, this assumption for droplet shape is widely applied in previous studies on droplet impacting solid substrate¹⁸⁻²² and works well in the theoretical models to predict droplet impact dynamics. Thus, despite the difference from reality, the deviation is small if the readers check the snapshots at 4.2 ms (maximum spreading) in Figure 1(a-c). Thus, we believe the cylindrical disk shape assumption is reasonable for the theoretical model in this study.

Referee #2 (Remarks to the Author):

The article, "Water sculpting effect:.....liquid film" demonstrates- according to the authors, a striking new phenomenon. However, the non-sticky nature of the oxide in the presence of the water/slip layer has been demonstrated several times in the literature. The current article, therefore, has insignificant novelties to be considered for Nat. Comm.

Literature survey: There are several articles published by Dickey Group (NCSU), Dickey in collaboration with Majidi (CMU), Shiyang Tang (UK), Khouroush (Australia), and Jing Liu (China), where the fluid-liquid metal interactions were explained in several ways. The influence of water on Gallium alloys was also dissected utilizing the contact angle and residual adhesion in one paper published in ACS AMI. The jumping liquid metal droplet work published by Dr. Dickey has phenomena similar to the current article. In fact, an oxidized metal droplet is hydrophilic. In the presence of water, the surface oxide changes. The low tension of the new oxide allows water to wet the adjacent surface. Therefore, it is quite intuitive to see the sculpting effect. Overall, this work seems fit elsewhere.

Response: We thank the referee for the time spent on our paper, constructive comments, and remarks. We are very pleased that the referee found our finding is "a striking new phenomenon." Regarding the referee's concern about the novelty of our manuscript compared to the previous literature, we have provided a detailed summary of previous studies related to our work, and highlighted their differences from our study as well as the novelty and new contributions of this study. We have also revised our manuscript to highlight our novelty from Lines 97 to 100. Meanwhile, we conducted extra experiments, polished our theoretical explanations and model, and carefully revised the paper following the referee's comments in the revised manuscript. In conclusion, we revealed the complete rebound of LM droplets originates from the trapping of a thin water film underneath that prevents droplet-solid contact, and the restitution coefficient is modulated by the negative capillary pressure as a result of the spontaneous spreading of water on the LM droplets. Based on these insights, we have developed a new theoretical model that is also in good agreement with our experimental results. We believe that these revisions and theoretical analyses enhance the novelty and scientific merit of our paper.

We have made a comprehensive investigation of the published papers related to the liquid

metal-interface interaction, including the papers from Dickey Group, Majidi (CMU), Shiyang Tang (UK), Khouroush (Australia), and Jing Liu (China). Among them, we found four papers that study the influence of water on the mobilization of Gallium alloys liquid metal (LM) ¹²³⁻²⁶, as summarized in Table R1. Specifically,

1) Khan et al. (2014) reported an underwater LM interfacial behavior in that the water provides a slip layer between the LM and solid surface, which increases the mobility of LM at a capillary tube. The oxide skin of the LM in water is also different from that in air, which weakens its mechanical strength. This study focuses on the underwater environment, but our study focuses on the atmosphere environment (a more general and a different scenario) where the LM surface is already oxidized by the air and forms a sticky oxidation skin showing high adhesion to the solid surface. Meanwhile, we studied the impact process where the drop inertia is essential for its dynamics which is highly different from Khan et al. (2014), and we highlighted that a thin water layer could help the LM droplet bounce from the solid surface due to the trapping of a thin water film underneath that prevents droplet-solid contact.

2) Song et al. (2021) found that an LM droplet at a Cu wire underwater (in NaOH solution) can jump off by using electrochemistry. This is because the switch of the potential applied to the metal drives electrochemical surface oxidation that changes the interfacial tension and the droplet shape and bounces the droplet off by transferring the interfacial energy to kinetic energy. Thus, the mechanism of drop bouncing is due to an electrochemical reaction (instead of the water) and is highly dependent on the environment, which is significantly different from our study.

3) Ding et al. (2016) reported that a water-coated LM droplet can disintegrate, merge, and even rebound after impacting the solid surface. Though the bouncing phenomenon is similar to our shell-core cases, they did not investigate the pure LM droplet impacting on a water layer-covered surface without systematic and deep investigation of the impact dynamics, including restitution coefficient. More importantly, they proposed that the behind physics of water-coated LM droplets bouncing as the oxidation of the liquid metal surface is different from that in our new proposed theoretical model.

4) Wang et al. (2016) found that the LM droplets impacting on a surface with micro/nano needle forest can rebound off due to the energy stored in the bent needle during impact. Clearly,

there is no water film in this study, and the LM droplet bouncing comes from a completely different mechanism.

Table R1. Summary of literatures related to LM droplet-interface interaction.

Paper/Author	Main conclusions
Khan, Mohammad R, ACS AMI, 2014 (Dickey's Group)	Discovery: This work shows that water can provide an interfacial slip layer between EGaIn and other surfaces, which allows the metal to flow smoothly through capillaries and across surfaces without sticking. Mechanism: Water, prior to adding metal, creates an interfacial slip layer between the surface of the metal plug and the walls of the microchannel that prevents the oxide from sticking to the walls. Difference from this study: This work mainly proves the importance of water film to liquid metal sliding, but the in-depth research on specific mechanisms, including slip or impact process, is not enough
Song M, Applied Physics Letters, 2021 (Dickey's Group)	Discovery: This work demonstrates an approach to get a single droplet of liquid metal (eutectic gallium indium) to jump by using electrochemistry in a solution of 1M NaOH. Mechanism: This rapid change in interfacial tension in the flattened state generates excess surface energy, which drives the droplet to return to a spherical shape with enough momentum that the liquid droplet jumps Differences from this study: This method has a limitation relative to conventional jumping drops is the need for electrolytes and a source of electricity to enable jumping.
Ding Y, Frontiers in Energy. 2016 (Jing Liu's Group)	Discovery: By encapsulating the liquid metal into water droplets, the fabricated liquid marble successfully avoided being oxygenized by the metal fluid. Such marbles could disintegrate, merge, and even rebound when impacting the substrate, unlike the existing dynamic fluidic behaviors of liquid marble or metal droplets. Mechanism: The oxidation of the liquid metal surface was efficiently prevented by the water coating, thus generating spherical marbles through free falling. Differences from this study: This work just preliminarily revealed the phenomenon of the water film also working as a lubricant, lacking a systematically investigating of the mechanisms lying behind these features.
Wang L, Advanced Materials Interfaces. 2016 (Jing Liu's Group)	Discovery: Bouncing LM droplets on micro/nano needle forest can rebound off the surfaces depending on the superhydrophobicity of the structure. Mechanism: The flexible needles can bend under the gravity of LM droplet and store energy, i.e., the kinetic energy of LM segmentally transfers into the elastic energy of the needles over their bending process. The stored energy in needles then turns to drive LM droplet to rebound off the surface in the recovery process. Differences from this study: Such rebound behavior relies on the complex regulation of the impacting surface, making micro/nanostructured surfaces with needle forest.

After above investigations, we summarize the novelty and new contributions of our study mainly from the following two aspects:

1) **New phenomena.** The novel phenomenon is characterized by the complete rebound of LM droplets from the water layer with negligible adhesion. More counterintuitively, the restitution coefficient, defined as the ratio between the droplet velocities after and before

impact, increases with water layer thickness. This novel phenomenon is of interest in various practical applications including heat dissipation, flexible circuit fabrication, and 3D-printed robot for enhanced jumping ability and mobility, as demonstrated in Fig. 4.

2) **New mechanisms.** The mechanism lies in the complete rebound of LM droplets originates from the trapping of a thin water film underneath that prevents droplet-solid contact, and the restitution coefficient is modulated by the negative capillary pressure as a result of the spontaneous spreading of water on the LM droplets. Considering the energy transformation, we have proposed a theoretical model which accurately predicts our experimental observations. These findings provide new understandings on the dynamics of the droplet impacting liquid-covered surfaces and theoretical guidance for the related practical applications and designs. Overall, we believe that the counterintuitive phenomena and the behind physics are novel enough and exhibit broad impact in multiple areas for it to be published in Nature Communications.

Referee #3 (Remarks to the Author):

In this paper the authors consider liquid metal droplets that rebound (jump) upon impact on a prewetted surface. The authors provide experimental results and propose a physical mechanism leading to the observed phenomenon. The mechanism is based essentially on a modification of surface tension of the impacting metal; such modification depends on the thickness of the prewetted layer, leading to larger jumps for the thicker ones.

The results are quite interesting, and as the authors point out, may be relevant not only as a peculiar fluid mechanical phenomena, but also in applications. One of these (walking robot) is outlined in the manuscript. So in principle I could be encouraging to move ahead with the reviewing process, however, the authors should improve the manuscript and their results first. Some rather crucial points are confusing at this point and these need to be clarified not to confuse the reader. I'll list my comments in a random order below - in my opinion, all of these should be addressed carefully before manuscript could be considered for publication.

Response: We thank the referee for the time spent on our paper, the complimentary comments, and the remarks. We are pleased to read that the referee found our "results are quite interesting" and suggested "move ahead with the reviewing process." We conducted extra experiments, polished our theoretical explanations and model, and carefully revised the paper according to the referee's comments. We believe that the scientific merit of our paper is now much improved. The details of the changes that we introduced are shown in the following responses, and changes to the text are highlighted in the text body.

Comments 1: The presentation: this needs some work - sculpting effect (where is this coming from?); page 2 - 'magic knife' - I don't think there is any magic here, please use less flowery language; 'sculpts the sticky skin' - well now, in addition to 'sculpts' we need to explain 'sticky' and 'skin'. I think I know what the authors mean, but this type of wording (and lack of justification) is not appropriate. The coefficient of restitution is -not- proportional to the water thickness as specified in the abstract - please be more precise.

Response: We thank the referee's helpful suggestion. We have carefully checked our manuscript, removed the flowery language, and used more precise and technical words in the revised manuscript. Specifically, we have deleted the "sculpting effect" in the title. We have

also deleted the phrase "magic knife." We have also deleted the words "sticky" in the title to avoid confusion. Now, the title of our paper is more precise as "Liquid metal droplets bouncing higher on thicker water layer." We deleted the word "skin" and "sticky" in the manuscript to make it more understandable.

Comments 2: The length scale - why using h_{LM} - thickness at max drop size - as the length scale? This one is different for each experimental condition, and I'm not sure the corresponding values are specified. Therefore the reader cannot bring the results back to the dimensional form. This is a serious problem.

Response: We thank the referee for the helpful comment. In our experiments, h_{LM} remains the same at a fixed We (Figure R2), so we used the droplet thickness h_{LM} as a characteristic length scale to non-dimensionalize the liquid film thickness h . However, as the concern arising from the referee, we realized that h_{LM} is not an initial parameter of the system and may not be appropriate to act as a characteristic length scale. Thus, we used the initial parameter of the system, the droplet radius R , as the characteristic length scale in the dimensionless forms in the revised manuscript, shown in Lines 103-104. For example, the dimensionless water layer thickness $H = h/R$, and the dimensionless maximum droplet diameter $R_m^* = R_m/R$. By doing this, the readers can easily bring the results back to the dimensional form.

Comments 3: What happens for even thicker water films? Do drops still rebound? What happens in the limit where water film thickness is orders of magnitude larger?

Response: We thank the referee for this insightful comment. Yes, the drop rebounds at even thicker water films, with a thickness of up to 10 mm (one order of magnitude larger than the maximum thickness in our manuscript), according to our supplementary experiments. As shown in Figure R5 (also see Supplementary Figure S1a), after release, the droplet penetrates the water surface and completely submerges into the water, then impacts the gas-liquid surface and finally bounces off. We can easily expect that this scenario is different from that in our manuscript since the effects of liquid-gas and liquid-solid surfaces are decoupled when the water layer is not a microlayer and the drop actually impact on a liquid-solid interface. It is an interesting topic that droplet can bounce off a solid surface in an underwater environment with

wide applications, which will be our future focus. However, this paper focuses on the situation where the water layer is thin enough to be considered as a microlayer (less than 1 mm) and the drop actually impact on a liquid-infused surface or a slippery surface.

To clarify the above points, we have added the claim in Lines 65-66 that "We focus on the drop impact at a solid surface covered by a water microlayer, thus varying the layer thickness between 20 – 1000 μm " in the revised manuscript. We have also mentioned the experimental results of drop impact on a much thicker water layer-covered solid surface in Lines 76-78 in the revised manuscript and SI for the reader's reference.

Figure R5 Experimental snapshots of LM droplet bouncing off a water layer-covered glass slide with the water layer thickness $h = 10 \text{ mm}$. $We = 10.7$, $H = 6.7$. The droplet radius $R = 1.5 \text{ mm}$. The scale bar is 1mm.

Comments 4: Page 5 says that experimental observation for the critical water film thickness is 226.2nm. I do not believe that the authors are able to measure this quantity so precisely. What are the error bars?

Response: We thank the referee for this comment. We noted that there might be a typo in this comment – it should be 226.2 μm instead of 226.2 nm. We used the mobile platform of the microscope (QuantaTM 450; FEG) to measure the thickness of the water film. And the measurement error of the mobile platform is 1 μm .

We used the probe method was used to measure the thickness of water layer. On two mobile microscope platforms (QuantaTM 450; FEG), the superhydrophilic glass slide and the hydrophilic hanging probe were fixed horizontally and vertically. We first recorded the vertical position of the hanging probe, of which the tip just touched the glass slide. Then, we removed the probe vertically and wet the superhydrophilic glass slide with a water droplet to form the required thickness of the water layer. We recorded the vertical position of the hanging probe

again after the hanging probe slowly moved down until its tip touched the water layer surface, forming a meniscus on the tip^{27,28}. We could get the water layer thickness by such two position records. The measurement error is $1\mu\text{m}$. Since the critical water layer thickness is an estimation when e reaches a plateau in this study, we have modified the value of the critical dimensionless water layer thickness as $Hc \approx 0.15$ in the revised manuscript at Lines 105-109.

Comments 5: The fitting parameter alpha on page 5 comes as a surprise. Why is it needed and where is it coming from?

Response: We thank the referee for this insightful comment. In our previous manuscript, the fitting parameter α is used considering that the droplet shape at the maximum jump height may deviate from the spherical shape. $\alpha = 1.08$ in our previous model, which means $\alpha \approx 1$ (no correction), and the droplet can be safely assumed as a sphere. In the revised manuscript, we have revised the theoretical model according to the referee's suggestion, where the drop shape at maximum jumping height is regarded as spherical, and no fitting parameters are adopted in its surface energy term at maximum jumping height.

Comments 6: References. The author do not cite the papers considering jumping metal drops in the context where such jumping is caused by the initial conditions. While such context is different from the present one, it is relevant, since it provides a useful example of another setup where jumping droplets are possible. See Science 309, 2043 (2005) and PRL 111, 034501 (2013).

Response: Thanks for the referee's suggestions. The suggested literature is very helpful to our study. We have cited these papers in the Introduction.

Reference:

- 1 Mani, M., Mandre, S. & Brenner, M. P. Events before droplet splashing on a solid surface. *Journal of Fluid Mechanics* **647**, 163-185, doi:10.1017/s0022112009993594 (2010).
- 2 De Ruiter, J., Lagraauw, R., Van Den Ende, D. & Mugele, F. Wettability-independent bouncing on flat surfaces mediated by thin air films. *Nat. Phys.* **11**, 48-53, doi:10.1038/nphys3145 (2015).
- 3 Blanken, N., Saleem, M. S., Antonini, C. & Thoraval, M.-J. Rebound of self-lubricating compound drops. *Science advances* **6**, eaay3499 (2020).
- 4 Angarska, J. K. *et al.* Detection of the Hydrophobic Surface Force in Foam Films by Measurements of the Critical Thickness of the Film Rupture. *Langmuir* **20**, 1799-1806, doi:10.1021/la0357514 (2004).
- 5 Yoon, R.-H. & Jordan, J. The critical rupture thickness of thin water films on hydrophobic surfaces. *Journal of colloid and interface science* **146**, 565-572 (1991).
- 6 Wang, X. *et al.* Dynamics for droplet-based electricity generators. *Nano Energy* **80**, 105558, doi:10.1016/j.nanoen.2020.105558 (2021).
- 7 Fay, J. A. 53-63 (Springer US, 1969).
- 8 Ji, B., Yang, Z. & Feng, J. Compound jetting from bubble bursting at an air-oil-water interface. *Nat. Commun.* **12**, doi:10.1038/s41467-021-26382-w (2021).
- 9 Zavabeti, A. *et al.* A liquid metal reaction environment for the room-temperature synthesis of atomically thin metal oxides. *Science* **358**, 332-335, doi:10.1126/science.aao4249 (2017).
- 10 Doudrick, K. *et al.* Different shades of oxide: From nanoscale wetting mechanisms to contact printing of gallium-based liquid metals. *Langmuir* **30**, 6867-6877, doi:10.1021/la5012023 (2014).
- 11 Creighton, M. A. *et al.* Oxidation of Gallium-based Liquid Metal Alloys by Water. *Langmuir* **36**, 12933-12941, doi:10.1021/acs.langmuir.0c02086 (2020).
- 12 Xu, Q., Oudalov, N., Guo, Q., Jaeger, H. M. & Brown, E. Effect of oxidation on the mechanical properties of liquid gallium and eutectic gallium-indium. *Phys. Fluids* **24**, 063101, doi:10.1063/1.4724313 (2012).
- 13 Handschuh-Wang, S., Gan, T., Wang, T., Stadler, F. J. & Zhou, X. Surface tension of the oxide skin of gallium-based liquid metals. *Langmuir* **37**, 9017-9025, doi:10.1021/acs.langmuir.1c00966 (2021).
- 14 Yunusa, M., Amador, G. J., Drotlef, D.-M. & Sitti, M. Wrinkling Instability and Adhesion of a Highly Bendable Gallium Oxide Nanofilm Encapsulating a Liquid-Gallium Droplet. *Nano Lett.* **18**, 2498-2504, doi:10.1021/acs.nanolett.8b00164 (2018).
- 15 Chiechi, R. C., Weiss, E. A., Dickey, M. D. & Whitesides, G. M. Eutectic Gallium-Indium (EGaIn): A Moldable Liquid Metal for Electrical Characterization of Self-Assembled Monolayers. *Angew. Chem., Int. Ed.* **47**, 142-144, doi:10.1002/anie.200703642 (2008).
- 16 Handschuh-Wang, S., Chen, Y., Zhu, L. & Zhou, X. Analysis and Transformations of Room-Temperature Liquid Metal Interfaces - A Closer Look through Interfacial Tension. *ChemPhysChem* **19**, 1584-1592, doi:10.1002/cphc.201800129 (2018).
- 17 Dickey, M. D. *et al.* Eutectic gallium-indium (EGaIn): A liquid metal alloy for the

- formation of stable structures in microchannels at room temperature. *Adv. Funct. Mater.* **18**, 1097-1104, doi:10.1002/adfm.200701216 (2008).
- 18 Richard, D., Clanet, C. & Quéré, D. Contact time of a bouncing drop. *Nature* **417**, 811-811, doi:10.1038/417811a (2002).
- 19 Wildeman, S., Visser, C. W., Sun, C. & Lohse, D. On the spreading of impacting drops. *Journal of Fluid Mechanics* **805**, 636-655, doi:10.1017/jfm.2016.584 (2016).
- 20 Clanet, C., Béguin, C., Richard, D. & Quéré, D. Maximal deformation of an impacting drop. *Journal of Fluid Mechanics* **517**, 199-208, doi:10.1017/s0022112004000904 (2004).
- 21 Fedorchenko, A. I., Wang, A.-B. & Wang, Y.-H. Effect of capillary and viscous forces on spreading of a liquid drop impinging on a solid surface. *Phys. Fluids* **17**, 093104, doi:10.1063/1.2038367 (2005).
- 22 Collings, E. W., Markworth, A. J., McCoy, J. K. & Saunders, J. H. Splat-quench solidification of freely falling liquid-metal drops by impact on a planar substrate. *Journal of Materials Science* **25**, 3677-3682, doi:10.1007/bf00575404 (1990).
- 23 Khan, M. R., Trlica, C., So, J.-H., Valeri, M. & Dickey, M. D. Influence of Water on the Interfacial Behavior of Gallium Liquid Metal Alloys. *ACS Appl. Mater. Interfaces* **6**, 22467-22473, doi:10.1021/am506496u (2014).
- 24 Song, M., Mehrabian, N., Karuturi, S. & Dickey, M. D. Jumping liquid metal droplets controlled electrochemically. *Applied Physics Letters* **118**, 081601, doi:10.1063/5.0036416 (2021).
- 25 Ding, Y. & Liu, J. Water film coated composite liquid metal marble and its fluidic impact dynamics phenomenon. *Frontiers in energy*. **10**, 29-36, doi:10.1007/s11708-015-0388-0 (2016).
- 26 Wang, L., He, Z., Ding, Y., Zhou, X. & Liu, J. The Rebound Motion of Liquid Metal Droplet on Flexible Micro/Nano Needle Forest. *Adv. Mater. Interfaces* **3**, 1600008, doi:10.1002/admi.201600008 (2016).
- 27 Yum, K. & Yu, M.-F. Measurement of wetting properties of individual boron nitride nanotubes with the Wilhelmy method using a nanotube-based force sensor. *Nano Lett.* **6**, 329-333 (2006).
- 28 Wang, X. *et al.* Influence of head resistance force and viscous friction on dynamic contact angle measurement in Wilhelmy plate method. *Colloids and Surfaces A: Physicochemical and Engineering Aspects* **527**, 115-122 (2017).

REVIEWER COMMENTS

Reviewer #1 (Remarks to the Author):

The authors have changed the explanation of the phenomenon. Yet the model they propose has no solid proof and is self-contradicting. In the model, the Laplace pressure and the surface tension is believed to pull the liquid metal (LM) droplet downwards during the impact, which leads to the decrease of the restitution coefficient e when the water film is thin. If this model is true, then the core-shell droplet should also have a smaller restitution coefficient at the corresponding (dimensionless) water film thickness H . Because as the authors claimed, the time for water to fully engulf the LM droplet is on the order of 1 ms, much smaller than the contact time. Then for the core-shell droplet, the water encapsulating the LM droplet should also spread very fast on the substrate, thus producing a similar (if not the same) meniscus around the LM droplet. Then the restitution coefficient of the core-shell droplet should also be smaller than the reported value. Yet this is not the case.

The model has two major flaws. First, there is no proof that the LM droplet does not touch the substrate at all when a water film is present. This is essential since the metal-solid interface provides the largest adhesion, which is evident from the sticking droplet when there is no water film. In fact, it is hard to believe that the LM droplet does not touch the substrate at all even when the water film is very thin. In addition, when the We number is larger, the model does not fit the experimental results. Second, to assume the radius of curvature of the meniscus $r \sim h$ during the whole contact time is too strong an assumption. This partly explains why the model needs two fitting parameters.

The authors need to improve the model substantially before this manuscript can be considered as publishable for this journal.

Reviewer #3 (Remarks to the Author):

The authors have addressed the comments from the previous round carefully, and as a result improved the manuscript significantly. The results are interesting, they are well placed in context at this point, and the end comment about strider robots may be of interest to the readers. I don't have additional comment and can support consideration for publication.

Point-to-point Responses to Referees' Comments

Referee #1 (Remarks to the Author):

The authors have changed the explanation of the phenomenon. Yet the model they propose has no solid proof and is self-contradicting. In the model, the Laplace pressure and the surface tension is believed to pull the liquid metal (LM) droplet downwards during the impact, which leads to the decrease of the restitution coefficient e when the water film is thin. If this model is true, then the core-shell droplet should also have a smaller restitution coefficient at the corresponding (dimensionless) water film thickness H . Because as the authors claimed, the time for water to fully engulf the LM droplet is on the order of 1 ms, much smaller than the contact time. Then for the core-shell droplet, the water encapsulating the LM droplet should also spread very fast on the substrate, thus producing a similar (if not the same) meniscus around the LM droplet. Then the restitution coefficient of the core-shell droplet should also be smaller than the reported value. Yet this is not the case.

Response: We thank the referee for the time spent on our paper, constructive comments, and remarks. We realized that we did not present our results clearly and the referee thought that model we proposed was self-contradicting.

In our previous Figure 2(a), H represents the dimensionless water layer thickness (h/R) on the glass substrate for both core-shell droplet and bare LM droplet cases. However, in the core-shell droplet impact experiment, the effective water layer thickness during impact, which determines the Laplace pressure that pulls the LM droplet downwards, should be the sum of h and the thickness of the bottom water shell h_s . Thus, to avoid misunderstanding, we defined $H = h/R$ for the bare LM droplet impact and $H = (h + h_s)/R$ for the core-shell LM droplet impact. Now, it is easy to understand why the restitution coefficient e of the core-shell droplet in our previous manuscript reaches the plateau value of the e of bare LM droplet impact. In our previous manuscript, the maximum water shell thickness at the bottom of the LM droplet before impact is measured as $h_s = 313 \mu\text{m}$, yielding $H \geq 0.21 > H_c$, where $H_c = 0.15$ is the threshold of the H -independent e region for bare LM droplet impact. As shown in Figure R1(a) (the same as Figure 2a in the revised manuscript), using the newly defined H as horizontal coordinates, the experimental data of core-shell droplet impact agree well with that of the bare LM droplet impact, both following the theoretical model well (Figure R1b). This demonstrates that there is no contradiction between our experiments and the theoretical model.

Besides, to further validate our theoretical model, we have conducted extra experiments using a core-shell droplet with a smaller shell thickness $h_s = 55 \mu\text{m}$. As shown in Figure R1(a), the restitution coefficient e at a small H is smaller than that at a large H . The results of core-shell droplet impact agree well with that of the bare LM droplet impact, which is predicted well by our theoretical model. These results are consistent with the

referee’s claim that “the time for water to fully engulf the LM droplet is on the order of 1 ms, much shorter than the contact time. Then for the core-shell droplet, the water encapsulating the LM droplet should also spread very fast on the substrate, thus producing a similar (if not the same) meniscus around the LM droplet. Then the restitution coefficient of the core-shell droplet should also be smaller than the (previous) reported value”.

Figure R1. (a) The Restitution coefficient e as a function of dimensionless water layer thickness H for bare LM droplet ($H = h/R$) and core-shell LM droplet ($H = h + h_s/R$) impact on a water layer-covered solid surface. (b) Comparison between the experimental results and the theoretical prediction, showing a good agreement.

To clarify the above points, we have revised the definition of H for the core-shell droplets (Line 121), added the results of extra experiments, and revised Figures 2(a) and 3(b).

The model has two major flaws. First, there is no proof that the LM droplet does not touch the substrate at all when a water film is present. This is essential since the metal-solid interface provides the largest adhesion, which is evident from the sticking droplet when there is no water film. In fact, it is hard to believe that the LM droplet does not touch the substrate at all even when the water film is very thin. In addition, when the We number is larger, the model does not fit the experimental results.

Response: We thank the referee for this critical comment. Based on our experimental observations from bottom view on the LM drop impact (Figure 1d-e in the manuscript) and the theoretical argument on the minimum water layer thickness upon impact, we assert that the LM droplet does not touch the substrate when a water film is present in current experiments. Experimentally, the LM droplet sticks on the glass slide when there is no water film (Figure 1d in the manuscript), thus providing the largest adhesion which suppresses the drop bouncing, as claimed by the referee. However, when there is a thin water film, the evolution of the bottom view is largely different from that of no water film case, the LM droplet finally fully retracts from the glass slide, and no residuals are observed (14.7 ms in Figure 1e in the manuscript). Considering the strong

adhesion between the LM droplet and the glass substrate, we believe that the absence of residuals suggests that there is no direct contact between the LM droplet and the glass substrate. Meanwhile, we have also provided theoretical analysis to prove that there is no contact between the LM droplet and glass slides when there is a water layer. The minimum fluid film thickness between the impacting drop and the solid substrate is estimated as $h_m = 5ROh_f^{8/9}We_f^{-10/9}$, where $Oh_f = \mu_w/(\rho_m\gamma_{mw}R)^{0.5}$ and $We_f = \rho_mRu^2/\gamma_{mw}$ ¹⁻³. In this study, the minimum water layer thickness for LM droplet with $We = 10.7$ is calculated to be $\sim 0.37 \mu\text{m}$, larger than the critical rupture thickness of a water film ($< 0.1 \mu\text{m}$)^{4,5}, indicating that a continuous and stable water lubrication layer can be preserved during impact. Therefore, both our experimental observations and theoretical analysis well prove that the LM droplet does not touch the glass slide when there is a water layer.

To provide more experimental proof, we have conducted extra core-shell droplet impact experiments with the glass slide slightly inclined (with an inclination angle of 13.5°), and visualized the impact area under the microscope with high resolutions. Similar to the experiments of vertical impact in Figure 1(d, e) in the manuscript, the LM droplet sticks to the glass slide without bouncing when there is no water layer, and the stuck LM can be clearly observed under the microscope (Figures R2a and R2c). While the core-shell LM droplet bounces off the glass slide due to the presence of the water shell layer, and there is no residual of LM observed at the impact area under the microscope with high resolution (Figures R2b and R2d). This further proves that there is no contact between the LM droplet and the glass slide during impact with the presence of a thin water layer.

Figure R2. (a) Selected snapshots of a bare LM droplet impacting on an inclined glass surface and (b) a core-shell LM droplet impacting on an inclined surface with the thickness of the bottom shell $h_s = 55 \mu\text{m}$. The inclination angle of the glass slides is 13.5° and the scale bar is 1mm. (c) and (d) the microscope images of the impact area at the glass slide surface before (left) and after (right) impact in (a) and (b), respectively, with a magnification of 10X. The images clearly show that the LM droplet sticks to the glass slide upon impact without the presence of water (c), while there are no LM residuals on the glass slide, i.e., no contact between the LM droplet and the glass slide during impact with the presence of a thin water shell layer (d). The scale bar is 0.1 mm.

For the deviations between the experimental results and the theoretical predictions at large Weber numbers, we believe it is caused by the energy loss from the breakup of the droplet during retraction. As shown in Figure 3 with $We = 18.8$, a fast upward jet is generated during the retraction process, which breaks up soon and ejects a small droplet upwards. This phenomenon is also observed in the experiments with $We = 26.8$. This small but fast droplet causes extra energy loss, thus decreasing the restitution coefficient and increasing the scattering of the experimental data, as shown in Figure S2 (Supplementary Information).

Figure R3. Time sequences of a bare LM droplet impact on a water layer covered glass slide with $We = 18.76$ and $H = 0.25$, showing the droplet breaks up during retraction and ejects a small droplet. The scale bar is 1mm.

To clarify the above points, we have added our extra experiments of core-shell droplet impact inclined surfaces in the Supplementary Information and clarified it in the revised manuscript (Line 121). We have also added the explanations for the deviations at large Weber numbers in the caption of Figure S2 (Supplementary Information)..

Second, to assume the radius of curvature of the meniscus $r \sim h$ during the whole contact time is too strong an assumption. This partly explains why the model needs two fitting parameters.

Response: We thank the referee for this comment. For the assumption of the radius of curvature of the meniscus $r \sim h$ used in the model, here we provide a theoretical validation that refers to a previous study on an inviscid droplet spreading on a solid substrate by Bianche, Clanet & Quere ⁶. The droplet spreads on a solid surface rapidly ($t^{1/2}$ law) at the initial stage dominated by the inertia-capillary mechanism, and then spreads slowly ($t^{1/10}$ law) at the late stage dominated by visco-capillary mechanism. We estimate the radius of curvature as the spreading length of the water layer at the end of the initial spreading stage, $r \sim (\gamma l / \rho)^{1/4} T^{1/2}$, where l is the characteristic dimension of the spreading liquid supplier (in our experiment it should be the water layer thickness thus $l \sim h$) and T is the time duration of the initial spreading stage characterized by the inertia-capillary time, i.e., $T \sim (\rho h^3 / \gamma)^{1/2}$. Thus, we obtain $r \sim h$. Actually, it is reasonable, since h is the only characteristic length scale for the spreading of the thin water layer (much thinner than the droplet size) in the system.

The reason why we need two fitting parameters is because that the capillary force and surface tension force exerted on the impacting droplet vary due to the variation of the contact line length (due to droplet shape change), lift distance, and the meniscus inclination angle. For other energy terms in the theoretical model, there are no fitting parameters. We have clarified this point in the revised manuscript in Line 157: “*a and b are two fitting parameters considering the variation of the drop shape and lift distance and the meniscus inclination angle during retraction*”.

The authors need to improve the model substantially before this manuscript can be considered as publishable for this journal.

Response: We appreciate the referee for the insightful and helpful comments. We have carefully considered them and made corresponding revisions in the revised manuscript. Specifically, we have revised the figures using the newly defined water layer thickness to avoid confusion on the results of core-shell droplet impact. We have also conducted extra experiments to prove the consistency between core-shell droplet impact experiments and the theoretical model, and to prove no contact between the impacting droplet with the glass slide with the presence of a thin water layer. We believe that through comprehensive revisions, the theoretical model is now being reasonably validated and easier to understand. We also believe that with the help of the referee, the revised manuscript is suitable to be published in Nature Communications now.

Reference:

- 1 Mani, M., Mandre, S. & Brenner, M. P. Events before droplet splashing on a solid surface. *Journal of Fluid Mechanics* 647, 163-185, doi:10.1017/s0022112009993594 (2010).
- 2 De Ruiter, J., Lagraauw, R., Van Den Ende, D. & Mugele, F. Wettability-independent bouncing on flat surfaces mediated by thin air films. *Nat. Phys.* 11, 48-53, doi:10.1038/nphys3145 (2015).
- 3 Blanken, N., Saleem, M. S., Antonini, C. & Thoraval, M.-J. Rebound of self-lubricating compound drops. *Science advances* 6, eaay3499 (2020).
- 4 Angarska, J. K. et al. Detection of the Hydrophobic Surface Force in Foam Films by Measurements of the Critical Thickness of the Film Rupture. *Langmuir* 20, 1799-1806, doi:10.1021/la0357514 (2004).
- 5 Yoon, R.-H. & Yordan, J. The critical rupture thickness of thin water films on hydrophobic surfaces. *Journal of colloid and interface science* 146, 565-572 (1991).
- 6 Biance, A.-L., Clanet, C. & Quéré, D. First steps in the spreading of a liquid droplet. *Physical Review E* 69, doi:10.1103/physreve.69.016301 (2004).

REVIEWER COMMENTS

Reviewer #1 (Remarks to the Author):

The authors have changed the definition of the dimensionless water layer thickness H for core-shell droplets, thus Figure 2 is changed as well. They also performed additional experiments for the core-shell droplets with a smaller shell thickness. Now that the three groups of data collapse reasonably well. Yet how the water layer thickness is non-dimensionalized may need a second thought. Since water can spread on the substrate in a very short time, then maybe $(A_d/A_s) h_s$ should be used instead of h_s in the new definition of H . Here A_d is the surface area of the droplet, and A_s is the surface area of the substrate.

As for whether the LM droplet touches the substrate during impact, the authors' answer still does not clear the doubt. The authors claimed that the critical rupture thickness of a water film to be $0.1 \mu\text{m}$, citing reference Angarska et. al., Langmuir, 2004. However, the critical rupture thickness in this reference was that of an electrolyte solution (sodium dodecyl sulfate solution). Plus, the two surfaces of the film are very smooth. However, these two conditions are not met in the current paper. Obviously, there is no electrolyte in the water film. More importantly, the wrinkles at the bottom of the drop during the impact are very large. See Figure 1e, the scale bar is 1 mm , thus at 1.1 ms the size of the wrinkles is on the order of $100 \mu\text{m}$. This is larger than the initial water film thickness. It is highly unlikely that these wrinkles do not touch the substrate at all. No LM residual is left on the substrate when there is water film present cannot be a proof that the LM droplet does not touch the substrate. Several reasons can cause this. For example, it could be that the oxidation layer on the LM droplet is different when it is immersed in water than in air.

I suggest more experiments be performed. If it is too difficult to directly observe whether the LM droplet touches the substrate (for example, using Total Internal Reflection), then different kinds of liquid should be used to test the validity of the theory, i.e., Eq.(2) in the manuscript. I suggest to test at least one more different kind of liquid which has a different surface tension, because surface tension is one of the variables in Eq.(2). The authors can try silicone oil (of different viscosities), or kerosene (used to keep LM from oxidizing), or even ethanol, etc. Silicone oil has a smaller surface tension than water, it should also spread spontaneously on the substrate and on the LM droplet. If Eq.(2) is valid, then it should work for different liquids. Especially, it should be independent on the liquid viscosity, but only depend on the surface tensions of the liquid. Plus, if Eq.(2) works with different liquids with different liquids, it also means that surface tension is indeed the main reason that caused the change of the restitution coefficient.

Point-to-point Responses to Referees' Comments

Referee #1 (Remarks to the Author):

The authors have changed the definition of the dimensionless water layer thickness H for core-shell droplets, thus Figure 2 is changed as well. They also performed additional experiments for the core-shell droplets with a smaller shell thickness. Now that the three groups of data collapse reasonably well.

Response: We thank the referee for the time spent on our paper, constructive comments, and remarks. We are pleased that the referee found our data "collapse reasonably well." Empowered by the referee's insightful comments and suggestions, we have polished our explanation of the proposed theoretical model, and conducted extra experiments to support the theoretical model. We believe that our paper is now much improved. The details of the changes that we made are shown in the following responses, and changes to the text are highlighted in the text body.

Yet how the water layer thickness is non-dimensionalized may need a second thought. Since water can spread on the substrate in a very short time, then maybe $(A_d/A_s)h_s$ should be used instead of h_s in the new definition of H . Here A_d is the surface area of the droplet, and A_s is the surface area of the substrate.

Response: We thank the referee for this suggestion. We believe using h_s instead of $(A_d/A_s)h_s$ in the new definition of H for core-shell drop impact should be reasonable, because the spreading area of the water shell is much smaller than the whole glass slide (A_s) during impact. For a complete wetting droplet, it spontaneously spreads on a superhydrophilic surface at initial driven by surface tension and resisted by inertia, which denoted as the inertial spreading stage where the spreading radius R_s follows the $t^{0.5}$ power law¹. One can find that $R_s \sim R$ during the impact process (~ 10 ms) of the core-shell droplet we used¹. If we consider A_d as the spreading area of the water shell on the glass slide, we obtain $A_s \sim A_d$ and $(A_d/A_s)h_s \sim h_s$. This conclusion also holds for the case of droplet impact with initial inertia in our experiments, where $R_s = R_m \sim R$. Thus, we believe the spontaneous spreading of the water shell on the glass slide exerts an unnoticeable influence on the drop impact dynamics and the initial parameter h_s should be considered in the new definition of H .

As for whether the LM droplet touches the substrate during impact, the authors' answer still does

not clear the doubt. The authors claimed that the critical rupture thickness of a water film to be 0.1 μm , citing reference Angarska et. al., Langmuir, 2004. However, the critical rupture thickness in this reference was that of an electrolyte solution (sodium dodecyl sulfate solution). Plus, the two surfaces of the film are very smooth. However, these two conditions are not met in the current paper. Obviously, there is no electrolyte in the water film. More importantly, the wrinkles at the bottom of the drop during the impact are very large. See Figure 1e, the scale bar is 1 mm, thus at 1.1 ms the size of the wrinkles is on the order of 100 μm . This is larger than the initial water film thickness. It is highly unlikely that these wrinkles do not touch the substrate at all. No LM residual is left on the substrate when there is water film present cannot be a proof that the LM droplet does not touch the substrate. Several reasons can cause this. For example, it could be that the oxidation layer on the LM droplet is different when it is immersed in water than in air.

Response: We thank the referee for this insightful comment. It is a pity that we cannot use the direct interferometric method to confirm that the LM drop does not touch the glass slide, because the low reflectivity of the LM surface. However, we believe our theoretical argument regarding the critical rupture thickness of the water film and our experimental evidence of no LM residuals strongly suggest our claim of no contact between the LM and glass slide with the presence of a thin water film.

For the references of critical rupture thickness of the water film, we use the measurement results of Mahnke et al. 1999² in the revised manuscript, which reported a critical rupture thickness of 0.04 μm for a pure water film on a glass surface. Actually, most of the previous studies reported a rupture thickness of the aqueous liquid film of $\lesssim 0.1 \mu\text{m}^{2-6}$. They found that the addition of electrolytes (including the sodium dodecyl sulfate used in Angarska et. al. 2004³) increases the critical rupture thickness and thus stabilizes the liquid film. Though there are no literatures studying the rupture of water film exactly the same to our case, this is the most similar case that suggests the critical rupture thickness of the water film during LM impact is less than 0.1 μm to our best knowledge.

For the experimental evidence of no LM residuals with the presence of a water film, we note that the chemical reaction (\sim hours⁷) between the oxidation layer of LM and water is much slower compared to the drop impact dynamics (\sim 10 ms), indicating the LM oxidation layer should maintain the same properties after contact with the water layer during impact. The experimental results of Gan et al⁷. found that there is almost no gallium oxide hydroxide detected after the LM

droplet with an oxidation layer exposure to water for 2 hours. When the exposure time increases to 4 hours, the coverage of gallium oxide hydroxide is less than 5%. In contrast, the contact time of the LM droplet during impact in our experiments is about 10 ms. Thus, it is very likely that the surface properties of the LM droplet after impacting a water layer is same to that of bare LM droplet in our experiments. Since LM residuals are observed for bare LM droplet impacting bare glass slide, we believe that no LM residuals with the presence of a thin water layer indicates that no contact between the impacting LM droplet and glass slide.

In order to clarify the above points, we have added the citation supporting the critical rupture thickness of a pure water film in Line 98, and added more explanations to support the claim of no contact between LM and glass in Lines 92-94.

I suggest more experiments be performed. If it is too difficult to directly observe whether the LM droplet touches the substrate (for example, using Total Internal Reflection), then different kinds of liquid should be used to test the validity of the theory, i.e., Eq.(2) in the manuscript. I suggest to test at least one more different kind of liquid which has a different surface tension, because surface tension is one of the variables in Eq.(2). The authors can try silicone oil (of different viscosities), or kerosene (used to keep LM from oxidizing), or even ethanol, etc. Silicone oil has a smaller surface tension than water, it should also spread spontaneously on the substrate and on the LM droplet. If Eq.(2) is valid, then it should work for different liquids. Especially, it should be independent on the liquid viscosity, but only depend on the surface tensions of the liquid. Plus, if Eq.(2) works with different liquids, it also means that surface tension is indeed the main reason that caused the change of the restitution coefficient.

Response: We thank the referee for this helpful comment. In practical applications, liquid metal shows the potential of producing artificial swimmers and directional movement in water environment, so it's more general to investigate LM motion under water environment⁸. As suggested by the referee, in order to further validate our theoretical model and highlight the essential role of surface tension, we have conducted extra experiments using a surfactant aqueous solution layer of 3.5 mM/L sodium dodecylbenzenesulfonate. The surfactant aqueous layer has a smaller surface tension ($\gamma_{wa} = 0.031$ mN/m) and a smaller water-LM interfacial tension ($\gamma_{mw} = 0.484$ mN/m), compared to those of pure water layer case. Figure R1 presents selected snapshots of a LM droplet impacting on surfactant aqueous solution layer covered glass substrates with

different dimensionless film thickness of $H = 0.04$ and 0.2 , showing similar bouncing behaviors compared to those of pure water layer cases.

Figure R1. Selected snapshots of high-speed camera images showing the different impact patterns of LM droplet impinging on the surface with H of 0.04 (a) and 0.2 (b) The LM droplet radius $R = 1.5$ mm. The scale bar is 1mm. Here $We = 10.72$, $\gamma_{wa} = 0.031$ mN/m, $\gamma_{mw} = 0.484$ mN/m.

As shown in Fig. R2 (also see Supplementary Figure S3b), the restitution of the LM droplet e after impacting on a surfactant solution liquid layer increases with the dimensionless layer thickness first and then reaches a plateau of about 0.36 , which is similar to that of pure water layer case. Meanwhile, our theoretical model also predicts the surfactant case reasonably well. As indicated by the referee, our extra experiments suggest that our model works with different liquid layers, and prove that surface tension is indeed the main reason that causes the change of the restitution coefficient.

In order to further support the validation of our theoretical model, we have added our extra experimental results with the surfactant solution liquid layer in Supplementary Figure S3b, and clarified it in Lines 170-175 in the main text.

Figure R1. Restitution coefficient e as a function of dimensionless water layer thickness H for bare LM droplets impacting a glass slide covered by a surfactant aqueous liquid layer of 3.5 mM/L sodium dodecylbenzenesulfonate. The theoretical model agrees well with the experimental results. Here $We = 10.72$.

Reference:

- 1 Biance, A.-L., Clanet, C. & Quéré, D. First steps in the spreading of a liquid droplet. *Physical Review E* 69, doi:10.1103/physreve.69.016301 (2004).
- 2 Mahnke, J., Schulze, H., Stöckelhuber, K. & Radoev, B. Rupture of thin wetting films on hydrophobic surfaces: Part I: methylated glass surfaces. *Colloids and Surfaces A: Physicochemical and Engineering Aspects* 157, 1-9 (1999).
- 3 Angarska, J. K. et al. Detection of the Hydrophobic Surface Force in Foam Films by Measurements of the Critical Thickness of the Film Rupture. *Langmuir* 20, 1799-1806, doi:10.1021/la0357514 (2004).
- 4 Yoon, R.-H. & Jordan, J. The critical rupture thickness of thin water films on hydrophobic surfaces. *Journal of colloid and interface science* 146, 565-572 (1991).
- 5 Pugh, R. & Yoon, R. Hydrophobicity and rupture of thin aqueous films. *Journal of colloid and interface science* 163, 169-176 (1994).
- 6 Manev, E. D. & Nguyen, A. V. Critical thickness of microscopic thin liquid films. *Advances in colloid and interface science* 114, 133-146 (2005).
- 7 Gan, T., Handschuh-Wang, S., Shang, W. & Zhou, X. GaOOH Crystallite Growth on Liquid Metal Microdroplets in Water: Influence of the Local Environment. *Langmuir* 38, 14475-14484, doi:10.1021/acs.langmuir.2c02539 (2022).
- 8 Sheng, L., Zhang, J. & Liu, J. Diverse transformations of liquid metals between different morphologies. *Adv Mater* 26, 6036-6042, doi:10.1002/adma.201400843 (2014).

REVIEWER COMMENTS

Reviewer #1 (Remarks to the Author):

The authors have performed extra experiments with a surfactant solution. The experiments seems to fit with their theoretical model (Eq.1).

The authors again uses Ref. 2-6 to say that the critical film thickness could be less than 0.1 μm . However, the condition in this paper is completely different from that of Ref. 2-6, since none of their film is under an impacting condition. The authors are aware of this. Plus, in this paper the bottom surface of the LM drop during impacting is not smooth but rough, which the authors ignore again. I cannot see how Ref. 2-6 could prove that the LM drop does not touch the substrate at all. The conditions are vastly different.

Actually, there is another way to bypass the problem that whether the drop touches the substrate, that is to test other liquids of different surface tension and different viscosity. If Eq. 1 holds for liquids of different surface tension and different viscosity, then it is less important that whether the LM drop touches the substrate. The authors have performed extra experiments with a surfactant solution, the results of which fits with the theoretical model. However, the surfactant solution has more or less the same viscosity as water, thus it remains a problem that in what viscosity range is Eq.1 applicable. Since viscosity does not enter into Eq.1, it suggests that Eq.1 should be applicable to liquids of any viscosity, which is questionable. Combined with the fact that it is not proved that the LM drop does not touch the substrate at all, then there is a possibility that Eq.1 is only a lucky guess which only holds for water or surfactant laden water. The two fitting parameters in Eq.1 might cover up the possible other mechanisms. Thus, in my view, it is dangerous to accept the paper in its current form since the validity of Eq.1 is still questionable. It will be best if the authors could perform another set of experiments with another pure liquid of a different viscosity, to prove that Eq.1 is applicable to liquids of different types -- both polar and non-polar liquids -- and viscosities.

Reviewer #4 (Remarks to the Author):

This manuscript presents experimental observations and an analytic model of the bouncing of a liquid metal droplet upon impact with a solid substrate covered by a thin fluid layer. Droplets are observed from below and from the side using a high-speed camera that allows detailed characterization of their motion. The analytic model is built by asserting that energy must be conserved from the moment during impact when the drop has spread maximally, to the moment when the drop reaches its maximum bounce height. Discussing potential applications to robotics, the manuscript also shows that a small

plastic object bounces higher and slides more freely with liquid metal droplets attached to its feet than without.

The manuscript offers evidently high-quality data, clear figures, and lucid prose. However, it presents relatively few results, the most important being the close agreement between analytic predictions and measurements of the coefficient of restitution, which holds for only a fairly narrow range of cases.

At minimum, the manuscript should include more discussion of the caveats and limitations of the model. Using conservation of energy to predict the maximum bounce height, as is done in Eq. 1, is a great idea. However, viscosity does not appear in the equation, and viscous effects altogether neglected. They are not zero; viscosity certainly turns some kinetic energy into heat as the droplet and its shell change shape during the interval from its moment of maximum spread to its moment of maximum height. The authors are right to quantify the importance of viscosity with Ohnesorge numbers -- one for the liquid metal (Oh_m) and one for the shell (Oh_w). The authors correctly identify viscous effects as negligible for the specific case of gallium-indium droplets coated with water, in which $Oh_m \ll 1$ and $Oh_w \ll 1$. In my own calculations, I found that $Oh_m \ll 1$ is also true for other common low-temperature liquid metals (4.9e-4 for Hg, 1.0e-3 for Li, 1.3e-3 for Na). If similar experiments were performed using one of these fluids instead of gallium-indium, the proposed model would again work well.

However, the $Oh_w \ll 1$ simplification fails for common fluids other than water, such as oils. If similar experiments were performed using viscous oil instead of water, the proposed model would lead to inaccurate predictions. The authors consider larger values of Oh_w when arguing that their results are surprising and counter-intuitive, but fail to point out that the model would fail when Oh_w is larger. The model would be far more powerful and the manuscript substantially improved if viscous effects were included. To do so would probably mean adding a viscous stress power term to Eq. 1. Approximations would be necessary because, in principle, stress power depends in detail on the evolution over time of the droplet's shape and internal velocity field. Perhaps something reasonable could be devised nonetheless with a few simplifications, helped along because the contact time is known from experiments.

An improved model could then be tested experimentally. The authors have experimented previously with water droplets bouncing on oil-infused surfaces, so why not liquid metal droplets, coated in oil, bouncing on glass slides coated with thin oil layers?

On the other hand, if viscosity cannot reasonably be handled in the model, the manuscript should be expanded with a paragraph discussing the range of applicability of the model, including examples of shell fluids and droplet sizes for which it is likely to be accurate and inaccurate.

I will close with a separate question: What is learned by determining the optimum values of the fit parameters a and b ? It seems to me that each term in Eq. 1 is the product of a surface energy and a surface area, so a and b can be interpreted as aspect ratios. Perhaps the manuscript could discuss the values and interpretation of a and b before concluding.

Point-to-point Responses to Referees' Comments

Referee #1 (Remarks to the Author)

The authors have performed extra experiments with a surfactant solution. The experiments seem to fit with their theoretical model (Eq.1).

Response: We thank the referee for the time spent on our paper, constructive comments, and remarks. Empowered by the referee's insightful comments and suggestions, we have validated the existence of a continuous water film by the observation of interference fringes during impact, and improved our theoretical model by considering viscous terms which agree well with our extra experiments using more viscous liquid layer. We believe that our paper is now much improved. The details of the changes that we made are shown in the following responses, and changes to the text are highlighted in the text body.

The authors again use Ref. 2-6 to say that the critical film thickness could be less than 0.1 μm . However, the condition in this paper is completely different from that of Ref. 2-6, since none of their film is under an impacting condition. The authors are aware of this. Plus, in this paper the bottom surface of the LM drop during impacting is not smooth but rough, which the authors ignore again. I cannot see how Ref. 2-6 could prove that the LM drop does not touch the substrate at all. The conditions are vastly different.

Response: We understand the concerns from the reviewer on the existence of a continuous water cushion in our experiments, which lacks experimental evidence in our previous manuscript. We have conducted experiments to image the interference fringe patterns using reflection interference contrast microscopy (RICM) coupled with a high-speed camera. An interference fringe is clearly observed underneath the LM droplet throughout the bouncing process, as shown in Figure. R1 (also see Figure. 1f in the revised manuscript), Figure. R2 (also see Supplementary Figure S3), and Supplementary Movie 3. This proves that an intact water cushion exactly exists between the impacting LM droplet and the glass slide, which prevents the LM droplet from contacting the glass slide. This thin lubrication film facilitates the bouncing of LM droplet. Our interference fringe observation is consistent with the experimental observation of no LM residuals on the glass surface after droplet bouncing (Figure. 1e and Supplementary Figure S2), and also consistent with our

theoretical analysis on the critical film thickness (Line 90-93 and 96-99).

We have noted that the interference fringes in our experiments is non-uniform, which is different from that of water droplet impacting oil-infused surface in our previous study (Hao et al. 2015)¹. This is because that the oxidation later at the LM surface shows some elastic properties during impact and generates wrinkles at the LM droplet surface during impact. Actually, the non-circular interference fringes we observed here is similar to that observed between the tissue culture cell and the glass slide, as reported by Chiu et al. (2012)². In conclusion, the inference fringes observed in the whole contact area during bouncing process prove the existence of a continuous water film that prevents the contact between LM and glass slide.

Figure R1. Interference fringe patterns imaged by the RICM confirm the existence of a continuous thin water layer during the impact process. The enlarged blue and red circled boxes clearly show the interference fringe patterns at the center and edge regions beneath the impacting LM droplet, respectively. The scale bars represent 1 mm.

Figure R2. Interference fringes imaged by the RICM for bare LM droplet impacting on water layer-covered glass slide. (a) $We = 10.72$, $H = 0.2$. (b) $We = 10.72$, $H = 0.06$. The scale bar is 0.2 mm.

Actually, there is another way to bypass the problem that whether the drop touches the substrate, that is to test other liquids of different surface tension and different viscosity. If Eq. 1 holds for liquids of different surface tension and different viscosity, then it is less important that whether the LM drop touches the substrate. The authors have performed extra experiments with a surfactant solution, the results of which fits with the theoretical model. However, the surfactant solution has more or less the same viscosity as water, thus it remains a problem that in what viscosity range is

Eq.1 applicable. Since viscosity does not enter into Eq.1, it suggests that Eq.1 should be applicable to liquids of any viscosity, which is questionable. Combined with the fact that it is not proved that the LM drop does not touch the substrate at all, then there is a possibility that Eq.1 is only a lucky guess which only holds for water or surfactant laden water. The two fitting parameters in Eq.1 might cover up the possible other mechanisms. Thus, in my view, it is dangerous to accept the paper in its current form since the validity of Eq.1 is still questionable. It will be best if the authors could perform another set of experiments with another pure liquid of a different viscosity, to prove that Eq.1 is applicable to liquids of different types -- both polar and non-polar liquids -- and viscosities.

Response: We thank the reviewer for the insightful comments. In our revised manuscript, we conducted interference experiments to prove that the LM drop actually does not touch the substrate with the existence of a water layer, which is explained in our last response. Furthermore, we have also conducted experiments with other liquids of different viscosities following the reviewer's suggestion, and the experimental results are well captured by our new derived theoretical model. To test the effect of the liquid layer viscosity, we adopted glycerin aqueous solutions with mass fractions of 10%-40% to adjust the liquid layer viscosity (corresponding to a viscosity of 1.3-3.7 mPa s) in the experiments. The LM droplet can successively bounce off the glass slide (Figure R3), and the restitution coefficient e shows the similar trend with the pure water layer case. We found that a higher liquid layer viscosity results in a smaller e , which indicates that the liquid viscosity affects the droplet bouncing behavior by changing the viscous dissipation in the liquid layer. Thus, we polished our theoretical model by adding viscous dissipation terms in the energy balance equation and obtained a new formula describing the restitution coefficient:

$$e = \left\{ \frac{1}{2} - C_1 \cdot \frac{3}{2We} \cdot \frac{\gamma_{wa}}{\gamma_{ma}} \cdot \frac{1}{H} - C_2 \cdot \frac{3\pi}{10} \cdot We^{10/9} Oh_f^{1/9} \left(\frac{\gamma_{mw}}{\gamma_{ma}} \right)^{-11/18} \right\}^{0.5},$$

Figure R3. Selected snapshots of high-speed camera images showing the bouncing of LM droplet impinging on glass slide covered by a 10 wt% glycerin aqueous solution layer. (a) $We = 10.7$, $H = 0.1$ (b) $We = 10.7$, $H = 0.15$. The scale bar is 1 mm.

where $Oh_f = \mu_w/(\rho_m \gamma_{mw} R)^{0.5}$ with μ_w representing the liquid layer viscosity. We compared the theoretical prediction with the experimental obtained e in the case of large H ($H > 0.2$) where e is independent of H , and found that our theoretical model captures the experimental results well, as shown in Figure R4 (also see Figure. 3c in the revised manuscript).

Figure R4. Theoretical predictions of e^2 as a function of Of_f ($=\mu_w/(\rho_m \gamma_{mw} R)^{0.5}$) using Eq. (3) in the revised manuscript agrees well with the experimental results at $H > 0.2$.

We believe the extra experiments using other liquid with different viscosities, combining with the experiments of interference fringes and the theoretical analysis and model in our revised manuscript are consistent with each other, and strongly prove that the LM droplet bouncing is caused by the existence of a thin water lubrication film and the restitution coefficient is modulated by the liquid layer thickness due to the capillary effects.

Reviewer #4 (Remarks to the Author)

This manuscript presents experimental observations and an analytic model of the bouncing of a liquid metal droplet upon impact with a solid substrate covered by a thin fluid layer. Droplets are observed from below and from the side using a high-speed camera that allows detailed characterization of their motion. The analytic model is built by asserting that energy must be conserved from the moment during impact when the drop has spread maximally, to the moment when the drop reaches its maximum bounce height. Discussing potential applications to robotics, the manuscript also shows that a small plastic object bounces higher and slides more freely with liquid metal droplets attached to its feet than without.

The manuscript offers evidently high-quality data, clear figures, and lucid prose. However, it presents relatively few results, the most important being the close agreement between analytic predictions and measurements of the coefficient of restitution, which holds for only a fairly narrow range of cases.

Response: We thank the referee for the time spent on our paper, constructive comments, and remarks. We are pleased that the reviewer found our manuscript offers “evidently high-quality data, clear figures, and lucid prose”. To extend the scope of the application of our findings, we have conducted extra experiments on LM droplet impact using other liquids with different viscosities to test the effect of liquid viscosity, and interference fringes observation to prove the existence of a continuous water cushion. Meanwhile, we have improved the theoretical model by including the viscous dissipations in LM droplet and the water film, which well describes the experimental results with different liquid viscosities. We believe that our paper is now much improved. The details of the changes that we made are shown in the following responses, and changes to the text are highlighted in the text body.

At minimum, the manuscript should include more discussion of the caveats and limitations of the model. Using conservation of energy to predict the maximum bounce height, as is done in Eq. 1, is a great idea. However, viscosity does not appear in the equation, and viscous effects altogether neglected. They are not zero; viscosity certainly turns some kinetic energy into heat as the droplet and its shell change shape during the interval from its moment of maximum spread to its moment of maximum height. The authors are right to quantify the importance of viscosity with Ohnesorge numbers -- one for the liquid metal (Oh_m) and one for the shell (Oh_w). The authors correctly identify

viscous effects as negligible for the specific case of gallium-indium droplets coated with water, in which $Oh_m \ll 1$ and $Oh_w \ll 1$. In my own calculations, I found that $Oh_m \ll 1$ is also true for other common low-temperature liquid metals (4.9e-4 for Hg, 1.0e-3 for Li, 1.3e-3 for Na). If similar experiments were performed using one of these fluids instead of gallium-indium, the proposed model would again work well.

However, the $Oh_w \ll 1$ simplification fails for common fluids other than water, such as oils. If similar experiments were performed using viscous oil instead of water, the proposed model would lead to inaccurate predictions. The authors consider larger values of Oh_w when arguing that their results are surprising and counter-intuitive, but fail to point out that the model would fail when Oh_w is larger. The model would be far more powerful and the manuscript substantially improved if viscous effects were included. To do so would probably mean adding a viscous stress power term to Eq. 1. Approximations would be necessary because, in principle, stress power depends in detail on the evolution over time of the droplet's shape and internal velocity field. Perhaps something reasonable could be devised nonetheless with a few simplifications, helped along because the contact time is known from experiments. An improved model could then be tested experimentally. The authors have experimented previously with water droplets bouncing on oil-infused surfaces, so why not liquid metal droplets, coated in oil, bouncing on glass slides coated with thin oil layers?

Response: We appreciate the reviewer for this insightful comment. We agree that our previous model does not account for the viscous effects in the LM droplet and liquid layer, and may fail to predict the cases with more viscous liquid layers. Inspired by the reviewer's suggestion, we have improved our model by adding the viscous terms ($E_{\mu m}$ and $E_{\mu w}$ in Eq. R1) and derived a new model considering the effect of liquid layer viscosity. We have also conducted LM drop impact experiments using more viscous liquids, and proved that the theoretical model well describes the experimental results.

First, we explain how we revised the theoretical model. Considering the energy conservation from the initial moment before falling to the maximum bouncing height (h_r) (Figure. R5a), we have

$$E_{p0} + E_{s0} - W_{Fc} - E_{\mu m} - E_{\mu w} = E_p + E_s, \quad (R1)$$

where $E_{p0} = mgh_0$ and $E_{s0} = 4\pi\gamma_m a R^2$ are the gravitational energy and surface energy at the initial moment before falling, respectively. $E_p = mgh_r$ and $E_s \approx E_{s0}$ are the gravitational energy and surface

energy at the maximum bouncing height, respectively, where $h_r = e^2 h_0$. W_{F_c} is the work done by F_c and expressed as $C_1 \pi R^2 \gamma_{wa} / h$ where C_1 is a fitting parameter. $E_{\mu m}$ and $E_{\mu w}$ are the viscous dissipations in the LM droplet and the beneath water film, respectively.

Figure R5. Theoretical modeling of water layer-induced drop bouncing. (a) Schematic drawing showing that the LM droplet falls from a height of h_0 above the glass slide at the initial moment, and then deforms as a pancake shape and suffers from a capillary suction force F_c during impact, and finally jumps to a maximum height of h_r . (b) Theoretical calculation of restitution coefficient e as a function of H with Eq. (R2) (Eq. 2 in the revised manuscript), which is in good agreement with the experimental results. (c) Theoretical predictions of e^2 as a function of Of_f ($=\mu_w/(\rho_m\gamma_{mw}R)^{0.5}$) using Eq. (R3) (Eq. 3 in the revised manuscript) agrees well with the experimental results at $H > 0.2$.

The existence of a continuous liquid cushion during impact facilitates a free-slip boundary condition. In this case, Wildmen et al. ³ found that the viscous dissipation in the impacting droplet

is approximately one-half of the initial kinetic energy when $We \geq O(10)$, thus we have $E_{\mu m} = 0.5E_{p0}$.

Based on the lubrication approximation theory, the energy dissipation in the water film can be estimated as $E_{\mu w} = C_2 h_w R^2 \mu_w (u/h_w)^2 \tau$.¹³ Here the water layer thickness h_w during impact is scaled using the minimum water layer thickness, i.e., $h_w \sim 5ROh_f^{8/9} We_f^{-10/9}$. $\tau \sim (\rho_m R^3 / \gamma_{ma})^{1/2}$ is the contact time. C_2 is a fitting parameter. Then, the droplet restitution coefficient e can be deduced from Eq. (R1) as

$$e = \left\{ \frac{1}{2} - C_1 \cdot \frac{3}{2We} \cdot \frac{\gamma_{wa}}{\gamma_{ma}} \cdot \frac{1}{H} - C_2 \cdot \frac{3\pi}{10} \cdot We^{10/9} Oh_f^{1/9} \left(\frac{\gamma_{mw}}{\gamma_{ma}} \right)^{-11/18} \right\}^{0.5} \quad (R2)$$

Here $Oh_f = \mu_w / (\rho_m \gamma_{mw} R)^{0.5}$ considering the effect of the liquid layer viscosity μ_w . Eq. (2) suggests that increasing the viscosity of the liquid layer decreases the restitution coefficient of the bouncing droplet by increasing the energy dissipation in the liquid layer, and there should be an upper limit for the liquid layer to completely inhibit the bouncing of LM droplet. Meanwhile, adopting $C_1 = 0.2$ and $C_2 = 0.535$, Eq. (R2) quantitatively agrees with our experimental results with water layer (Figure R5) and surfactant surface layers (Figure R6), validating the applicability of our model to capture the effect of surface tension^{4, 5}.

Figure R6. Restitution coefficient e as a function of dimensionless liquid layer thickness H for bare LM droplets impacting on a glass slide covered by a surfactant aqueous liquid layer of 3.5 mM/L sodium dodecylbenzenesulfonate. Here $We = 10.72$, $\gamma_{wa} = 0.031$ mN/m and $\gamma_{mw} = 0.484$ mN/m. The theoretical predictions using Eq. (R2) (Eq. 2 in the revised manuscript) agree well with the experimental results.

Next, we introduce our extra experiments using more viscous liquids and the comparison between experiments and theoretical model. To test the effect of the liquid layer viscosity, we adopt glycerin solutions with mass fractions of 10%-40% to adjust the liquid layer viscosity (corresponding to a viscosity of 1.3-3.7 mPa s) in the experiments. We use glycerin aqueous solutions instead of oils because we want to change the liquid layer viscosity while maintain its surface and interfacial tensions. The γ_{wa} and γ_{mw} of glycerin solutions of different concentrations are similar to those of water, while the γ_{wa} and γ_{mw} of silicone oils are much smaller than those of water. Experimental results shows that the LM droplet can successively bounce off the glass slide (Figure R3), and the restitution coefficient e shows the similar trend with the water layer cases. We found that a larger liquid layer viscosity results in a smaller e (Figure R5b), which is well aligned with the prediction of Eq. (R2). For a thicker enough liquid layer ($H > 0.2$), we neglect the term of W_{Fc} in Eq. (R1). Considering the nearly constant γ_{mw} for different glycerin solutions and $We = 10.7$, we obtain from Eq. (R2) that

$$e = \left\{ \frac{1}{2} - 0.92Oh_f^{1/9} \right\}^{0.5} \quad (\text{R3})$$

for large H ($H > 0.2$), which well predicts our experimental results, as shown in Figure. R5(c).

We cannot test other LMs suggested by the reviewer. Hg is toxic and we do not have credential to conduct experiments with it in our lab. Li and Na have strong activities and flammability, and easily reacts with water at room temperature and thus cannot be used for LM drop impacting aqueous liquid layer experiments.

Finally, we discuss the limitations of our theoretical model by comparing this study with our previous study using a highly viscous oil layer¹. Our previous work on water droplet impact on slippery oil-infused surfaces found that the water droplet sticks on the oil-infused surface at the same We ($We > 9$), in contrast to the observations in this study. This contrast comes from the different roles of viscous dissipation in these two scenarios. Our previous work adopted a highly viscous oil layer (viscosity ≥ 0.15 Pa s) with $Oh_f \geq O(0.1)$, indicating a significant role of viscous dissipation. Using Eq. (3) yields a negative e , which means the droplet cannot bouncing off. However, in this study, the aqueous liquid layer above the solid surface has a much smaller viscosity yielding $Oh_f \leq O(10^{-3})$, suggesting a minor role of viscous dissipation and thereby causing a positive e , making the LM droplet successively bounce off the glass slide. This indicates

that only less viscous liquid layer can help the LM droplet to bounce off the solid surface, while a highly viscous liquid layer exerts high viscous dissipation which inhibits the bouncing. There exists an upper limit of the liquid viscosity for the LM bouncing we reported here.

In summary, we believe the above theoretical analysis and extra experimental studies improve our manuscript substantially, and extend the applications of our findings and theoretical models. We have comprehensively revised the manuscript (Line 148-204) and we hope these revisions address the reviewer's concern.

On the other hand, if viscosity cannot reasonably be handled in the model, the manuscript should be expanded with a paragraph discussing the range of applicability of the model, including examples of shell fluids and droplet sizes for which it is likely to be accurate and inaccurate.

Response: We thank the reviewer for this suggestion. As we responded to the last comment, we have handled the viscosity problem experimentally and theoretically, and the revised theoretical model aligns well with the experimental results. Regarding the influence of the liquid layer (or shell fluids), we have tested the influence of liquid viscosity experimentally and well explained it using the theoretical model. The range of applicability of the model considering the liquid viscosity is also discussed (Line 183-192). We hope these discussions help the readers to understand the limitation of our study and inspire more interests on the LM impact in the future studies.

I will close with a separate question: What is learned by determining the optimum values of the fit parameters a and b ? It seems to me that each terms in Eq. 1 is the product of a surface energy and a surface area, so a and b can be interpreted as aspect ratios. Perhaps the manuscript could discuss the values and interpretation of a and b before concluding.

Response: We thank the reviewer for this helpful question. In our revised theoretical model, we have neglected the term related to the surface tension force F_s considering F_s is negligible compared to the capillary force F_c because $h \ll R$ for most of the experimental cases. We used two fitting parameters of C_1 and C_2 in the revised model.

C_1 is applied in the term of work done by F_c : $W_{F_c} = C_1 \pi R^2 \gamma_{wa} / h$. We adopted C_1 because the water meniscus and droplet shape change with time during droplet impact, and thus the radius of curvature of the water meniscus r and the acting distance of F_c also vary with time. In this case,

we can only obtain the scaling of r using the liquid layer thickness h , and the scaling of the acting distance of F_c using the drop radius R . Thus the expression of W_{F_c} need a constant fitting parameter C_1 for simplification. Our fitting gives $C_1 = 0.2$ which is close to $O(1)$, indicating the rationality of the scaling we adopted.

C_2 is applied in the term of viscous dissipation in the liquid layer: $E_{\mu w} = C_2 h_w R^2 \mu_w (u/h_w)^2 \tau$. We adopted C_2 because the viscous stress depends in detail on the evolution over time of the droplet's shape and internal velocity field, as claimed by the reviewer in the last comment. Meanwhile, the contact time τ is scale as the capillary-inertia time which is also need a proportional coefficient. Our fitting gives $C_2 = 0.55$ which is very close to 1, indicating the rationality of the theoretical model we proposed.

In order to clarify the above points, we have added explanations when introducing C_1 and C_2 in the revised manuscript. Line 156-157: “ C_1 is a fitting parameter considering that the radius of curvature of the water meniscus as well as F_c and the acting distance of F_c vary with time during impact”. Line 164-166: “ C_2 is a fitting parameter for simplification considering the time variations of the droplet shape and the velocity field in the liquid layer during impact”.

References:

1. Hao C, *et al.* Superhydrophobic-like tunable droplet bouncing on slippery liquid interfaces. *Nat Commun* **6**, 7986 (2015).
2. Chiu LD, Su L, Reichelt S, Amos WB. Use of a white light supercontinuum laser for confocal interference-reflection microscopy. *Journal of Microscopy* **246**, 153-159 (2012).
3. Wildeman S, Visser CW, Sun C, Lohse D. On the spreading of impacting drops. *Journal of Fluid Mechanics* **805**, 636-655 (2016).
4. De Ruiter J, Lagrauw R, Van Den Ende D, Mugele F. Wettability-independent bouncing on flat surfaces mediated by thin air films. *Nat Phys* **11**, 48-53 (2015).
5. Blanken N, Saleem MS, Antonini C, Thoraval M-J. Rebound of self-lubricating compound drops. *Science advances* **6**, eaay3499 (2020).

REVIEWERS' COMMENTS

Reviewer #1 (Remarks to the Author):

The authors used reflection interference to show that the water film indeed exist during the impact. They also performed more experiments to show the influence of the viscosity of the liquid layer, which fits with the theoretical model. The authors have answers all my concerns, thus I recommend the manuscript to be published in Nature Communications.

Reviewer #4 (Remarks to the Author):

I appreciate that the authors added viscosity to their analytic model and performed experiments with glycerin-water solutions to explore the role of viscosity. The results agree well with their model, at least for thick lubrication layers. The combined results now make a lot of sense and show that viscous damping in the lubrication layer plays a key role. Highly viscous layers extract lots of energy from the droplet because fluid motion is coupled over long distances, so the droplet must entrain a lot of the lubrication layer. Thin layers extract lots of energy from the droplet because shear stresses are high. These improvements to the model and corresponding experiments make the manuscript significantly stronger.

Thus, I generally support publication, provided the authors can address a few remaining comments:

Line 198: The manuscript states that small thickness h_f yields a negative value for the coefficient of restitution e , when actually it yields an imaginary value for e . Furthermore, Eq. 3 predicts that e becomes imaginary, implying droplets cannot bounce, when $Oh_f \geq 0.29$. The manuscript should state this value and include a sentence or two discussing it. Is it consistent with experimental observations?

The expression for the capillary work $W_{\{Fc\}}$ on line 157 appears to be incorrect. It would imply that the units are J/m, but the units should be J. Is a length scale missing?

Line 257: The Nikon Ti-s is an inverted microscope, not a camera.

The Ohnesorge number is never named, though Oh_f is used throughout. Readers would benefit if the name were mentioned at the first use of the symbol.

Finally, the manuscript would benefit from language editing throughout. There are many minor grammar errors.

Point-to-point Responses to Referees' Comments

Referee #1 (Remarks to the Author)

The authors used reflection interference to show that the water film indeed exist during the impact. They also performed more experiments to show the influence of the viscosity of the liquid layer, which fits with the theoretical model. The authors have answers all my concerns, thus I recommend the manuscript to be published in Nature Communications.

Response: We are grateful for your thorough review of our revised manuscript and appreciate your positive feedback. We are pleased to hear that we have adequately addressed all of your concerns and that you recommend the publication of our work in Nature Communications. Thank you once again for your time, effort, and expertise in reviewing our manuscript. We are grateful for your support throughout this process.

Reviewer #4 (Remarks to the Author)

I appreciate that the authors added viscosity to their analytic model and performed experiments with glycerin-water solutions to explore the role of viscosity. The results agree well with their model, at least for thick lubrication layers. The combined results now make a lot of sense and show that viscous damping in the lubrication layer plays a key role. Highly viscous layers extract lots of energy from the droplet because fluid motion is coupled over long distances, so the droplet must entrain a lot of the lubrication layer. Thin layers extract lots of energy from the droplet because shear stresses are high. These improvements to the model and corresponding experiments make the manuscript significantly stronger.

Response: We sincerely appreciate your positive feedback and acknowledgement of the improvements we made to our analytic model and experimental approach. We are glad that our addition of viscosity to the model and the corresponding experiments with glycerin-water solutions have strengthened the manuscript and provided valuable insights into the role of viscosity. Your recognition of these improvements and the overall strength of the manuscript is greatly appreciated. We are grateful for your thoughtful evaluation and support, which have undoubtedly contributed to the enhanced quality and significance of our work. Thank you once again for your time, expertise, and valuable feedback.

Line 198: The manuscript states that small thickness h_f yields a negative value for the coefficient of restitution e , when actually it yields an imaginary value for e . Furthermore, Eq. 3 predicts that e becomes imaginary, implying droplets cannot bounce, when $Oh_f \geq 0.29$. The manuscript should state this value and include a sentence or two discussing it. Is it consistent with experimental observations?

Response: We thank the reviewer for this insightful suggestion. We have stated this threshold value in Lines 186-190 in the revised manuscript: “When $Oh_f > 0.004$, the LM droplet cannot bounce off due to the large viscous dissipation in the liquid layer, yielding $e = 0$, which is validated by our experiments with a highly viscous aqueous liquid layer with $Oh_f = 0.0043$ (Supplementary Figure S8)”. This threshold value is consistent with our experimental observation, which shows that LM droplet impacting on a glass slide covered by a 60 wt% glycerin solution layer cannot bounce off, as we provided in Supplementary Figure S8.

The expression for the capillary work $W_{\{Fc\}}$ on line 157 appears to be incorrect. It would imply that the units are J/m, but the units should be J. Is a length scale missing?

Response: We thank the reviewer for pointing out this typo. We missed a length scale in the previous expression of W_{Fc} , which should be $C_1\pi R^3\gamma_{wa}/h$. We have revised it correspondingly in Line 151 as “ $W_{Fc} = C_1\pi R^3\gamma_{wa}/h$ ”.

Line 257: The Nikon Ti-s is an inverted microscope, not a camera.

Response: Thank you for bringing to our attention the incorrect labeling of the instruments used for capturing side-view and bottom-view images in the Methods section. In our study, we employed two different instruments to capture side-view and bottom-view images of the droplets during the experiments. Due to the labeling mistake, we incorrectly mentioned the instruments in the Methods section (Line 249).

The correct instruments are as follows:

Side-view images: We used a high-speed camera (Photron, Fastcam Nova S) to record the side-view motion of the droplets.

Bottom-view images: We utilized an inverted microscope (Nikon Ti-s) to capture bottom-view images of the droplets interacting with the substrate during the experiments.

The Ohnesorge number is never named, though Oh_f is used throughout. Readers would benefit if the name were mentioned at the first use of the symbol.

Response: Thank you for your valuable suggestion. We have named Oh_f and provided the physical meaning in Lines 93-95: “where $Oh_f = \mu_w / (\rho_m \gamma_{mw} R)^{0.5}$ is the Ohnesorge number that relates the viscous force to the inertial and surface tension forces”.

Finally, the manuscript would benefit from language editing throughout. There are many minor grammar errors.

Response: We have carefully checked the language typos and grammar errors along the manuscript and revised them accordingly. We believe now the manuscript reads much better.